# Can digital financial inclusion converge the regional agricultural carbon emissions intensity gap?

**Lingzhi Tan**[1]*, **Nuolan Tian**[2], **Xiaxuan Li**[2], **Huan Chen**[2]

**1** School of Public Affairs, Chongqing Technology and Business University, Chongqing, China, **2** School of Law and Sociology, Chongqing Technology and Business University, Chongqing, China

* wgytlz@126.com

## Abstract

To explore whether digital finance can reduce agricultural carbon emissions, promote regional convergence, and foster inclusivity in rural revitalization and shared prosperity, this paper uses the provincial-level index of digital financial inclusion to analyze the impact of digital financial inclusion on the intensity of agricultural carbon emissions and the Degum Gini coefficient (D-Gini coefficient) of regional carbon emission intensity in 30 sample provinces from 2010 to 2020. It examines the mechanism of the impact of digital financial inclusion on both variables to understand the underlying factors better. The main conclusions are as follows: (1) Digital financial inclusion significantly reduces the intensity of agricultural carbon emissions and narrows the gap in carbon emission intensity between regions. (2) The unconditional quantile regression coefficients show that the negative coefficients of the digital financial inclusion index and the three-dimensional indices decrease with increasing quantiles. However, the significant effects vary significantly at different quantiles. (3) Technological progress and the government's ability to allocate financial resources play a significant mediating role, and the income gap between urban and rural areas can be further narrowed, as well as the carbon emission intensity gap between provinces. The empirical results are robust and proven by replacing the econometric analysis method, changing the core variables, and other methods.

## 1. Introduction

Establishing a green, low-carbon, and sustainable global governance system has become a common development goal for all countries. Agricultural production is strongly affected by and a significant contributor to climate change. Agriculture and land-use change account for a quarter of total global emissions of greenhouse gases (GHG) [1]. In 20 years, China was responsible for the most emissions from agricultural production [2]. Consequently, developing low-carbon agriculture (LCA) with "high efficiency, low energy consumption, low emissions, and high carbon sinks" is crucial. Since the 18th Party Congress, China has continuously optimized its agricultural industry structure, resulting in a preliminary low-carbon agricultural

**Data Availability Statement:** Data cannot be shared publicly as metadata is not publicly available. Data is available from the Institute of Digital Finance, Peking University for researchers

who meet the criteria for accessing confidential data.

**Funding:** This work was supported by the Chongqing Philosophy and Social Science Project to LZT (2021NDYB087); The National Social Science Fund of China to CZW(22CGL034). The funders of the Chongqing Philosophy and Social Science Project to LZT was involved in the research design, data collection and analysis, and decision to publish or prepare the manuscript. The funders of The National Social Science Fund of China to CZW had no role in study design, data collection and analysis, decision to publish, or preparation of the manuscript.

**Competing interests:** The authors have declared that no competing interests exist.

development pattern. However, due to differences in the foundation of agricultural economic development and resource endowment between regions, achieving low-carbon development in all areas is still challenging. Agricultural carbon emissions' total amount and intensity also show significant regional polarization. In October 2021, the Opinions on the Complete and Accurate Implementation of the New Development Concept for Peak Carbon and Carbon Neutral Work explicitly proposed to promote carbon sequestration and efficiency in agriculture by accelerating the green development of agriculture. In the 20th Party Congress report, General Secretary Xi Jinping emphasized the importance of promoting coordinated regional development. The convergence of total and intensity differences in agricultural carbon emissions between regions is one manifestation of coordinated regional development [3].

Currently, the inefficient agricultural development model and the pattern of the smallholder economy remain the direct causal factors of high agricultural carbon emissions in China [4]. Reducing the intensity of agricultural carbon emissions requires the support of various social and economic capitals, especially sustained investment from various financial factors [5]. However, the agricultural sector often faces difficulties obtaining capital and resources, compounded by significant climate and market risks [5]. Traditional financial services and factor capital cannot proactively support agricultural carbon reduction [6, 7]. Financial exclusion, non-equilibrium, and volatility in financial markets further exacerbate the Matthew effect of regional agricultural carbon emission intensity [8].

The study of the impact of finance on provincial differences in carbon emissions has a long history. Many studies have analyzed this impact from a macro-regional perspective. For instance, Tamazian A. et al. (2009) confirmed that a sound financial system can promote regional economic growth and reduce carbon emission intensity [9]. Li W et al. (2023) examined the spatial impact of climate finance on mitigating the regional carbon emission gap [10]. Geng YQ (2021) analyzed the relationship between consumer behavior and the environment as a result of tourism [11]. Zhang F et al. (2022) confirmed that enterprise technological progress and innovation capacity are the important factors that affect provincial differences in carbon emission intensity [12]. High-level financial services play an essential role in promoting the green transformation of the economy, reducing the total regional carbon emissions, and other related aspects [13]. In particular, with the development of green finance, finance has received increasing attention as an essential factor affecting provincial differences in carbon emission intensity. For example, Wu J (2024) argues that green financial systems such as green credit, investment, and securities can encourage technological innovation and reduce regional heterogeneity in carbon emissions [14].

Developing sustainable digital technology is beneficial [15]. In recent years, digital financial inclusion has gained attention due to advancements in technological tools, such as cloud computing, the Internet, and big data. These tools have broken through traditional geographical and industry boundaries of financial factor flows, and research on the impact of digital financial inclusion has gradually developed. Early discussions on inclusive finance focused more on its economic and poverty reduction effects. However, about the accessibility and convenience of inclusive finance, research on the impact of regional carbon emissions has gradually been conducted. Chen M et al. (2022) confirmed significant provincial differences in carbon emissions due to digital financial inclusion based on county-level panel data [16].

Meanwhile, Doong A et al. (2022), Peng W (2022), and Zhang W et al. (2023) empirically tested the difference in agricultural carbon emission reduction between grain and non-grain major production areas [17–19]. Hu BT and Xiao YX (2022) found that the impact of digital financial inclusion on carbon emission reduction in eastern and less economically developed regions was more significant, and regional technological innovation had a significant mediating effect [20]. Some studies discuss carbon emissions as an influencing variable, such as He

MB and Yang XW (2021), who used carbon emissions as a mediating variable to confirm the bias of digital financial inclusion on total factor productivity in different regions [21]. Additionally, Zhang Y et al. (2022) found significant heterogeneity in the industry and regional impacts of inclusive finance on promoting green development in agriculture [22].

As discussed above, digital financial inclusion offers universality, inclusiveness, and a wide range of services, benefiting the "long-tail group" and promoting carbon emission reduction in agriculture. However, it has not been effectively verified whether the digital transformation of agriculture driven by inclusive finance can achieve the local dual carbon goals of agriculture, nor has it been demonstrating how inclusive finance can reduce the carbon emission intensity of adjacent regions through its convenience, cross-temporal and spatial characteristics, and low cost. Moreover, there is scarce evidence on how inclusive finance can achieve the convergence of agricultural carbon emissions among regions. To address these gaps, this paper uses panel data from 30 provinces (excluding Tibet and Hong Kong, Macao, and Taiwan) between 2010 and 2020, combined with Peking University's digital financial inclusion database, to verify the impact and mechanism of digital financial inclusion on the inter-regional agricultural carbon emission intensity gap.

## 2. Literature review and research hypotheses

### 2.1 The impact of digital financial inclusion on the carbon intensity of agriculture

China has continuously solved rural development problems in these decades and has obtained significant progress [23]. In the context of promoting shared prosperity and realizing rural revitalization, the development of the digital economy is expected to have a catalytic effect in breaking the barriers between urban and rural areas and reducing carbon emissions [24]. Digital financial inclusion, with its core of inclusiveness and extensiveness, not only improves financial efficiency but also provides environment-friendly financial services. It promotes the transformation of regional economic industries from high input and pollution to low energy consumption, high output, and green, low-carbon modes. It guides the transfer of resources, capital, and technology to low-carbon industries [25, 26]. Firstly, digital financial inclusion and technology can optimize financial resource allocation, enhancing resource use efficiency and laying the foundation for the green transformation of industries in rural areas. By leveraging information technology, big data, and artificial intelligence, traditional smallholder economic models and backward farming and production methods are gradually abandoned, and agricultural economy and resource utilization modes are adjusted. This effective transformation promotes upgrading industrial and consumption structures in rural areas [27, 28]. Farmers benefit from learning new skills and accessing more information through the digital economy platform, improving their human capital. With the support of digital financial inclusion, they have more opportunities for entrepreneurship and innovation, facilitating the low-carbon transformation of the agricultural industry [29]. Secondly, digital financial inclusion can provide a financial services platform for transforming the development approach of agriculture with carbon emission reduction as its core. Traditional finance faces the issue of diminishing marginal cost of financing and financial exclusion, where capital is mainly driven towards the non-agricultural sector, resulting in a lack of factors of production and capital required for agricultural development. This lack of resources and motivation discourages farmers from adopting new technologies, introducing new varieties, and developing new ideas, which exacerbates the high carbon emissions of agriculture.

Digital financial inclusion, with its low cost and convenience features, avoids the obstruction of the "last mile" of financial services in agriculture and rural areas [30–33]. It broadens

the source of funds and quantifies the needs of "long-tail groups" such as agriculture, rural areas, and farmers by accurately assessing demand. It also lowers the barriers to entry and transaction costs in the financial market. Driven by national policies, the development of low-carbon, green, and high-quality agriculture is being chosen by more people. Agricultural green transformation development receives financial support, leading to comprehensive and multi-source agricultural emissions reduction.

Furthermore, the demand for low-carbon development, driven by digital financial inclusion, can subconsciously raise the environmental awareness of the rural population. It actively chooses new technologies and production models for achieving local agricultural carbon emission reduction targets [34]. Based on these observations, this paper proposes the following hypotheses.

H1: Digital financial inclusion has an agricultural carbon emission reduction effect.

## 2.2 The provincial differences in agricultural carbon emissions and digital financial inclusion

Digital finance is not constrained by time or geographical space, making it more likely to expand faster than traditional finance. This rapid expansion enables the formation of financial networks on a larger scale, resulting in lower transaction costs and a more significant competitive advantage. Researchers believe that digital financial inclusion can promote the integration and development of finance and industry in different regions [20], significantly improving industrial productivity and resource utilization efficiency [35, 36]. Digital finance has less impact on income fluctuations and bankruptcy rates for SMEs in counties or rural areas than traditional finance, making it a more sustainable and stable option for promoting sustainable agricultural development. Consequently, digital financial inclusion can create a "multiplier effect" on agricultural carbon emission reduction [30]. The spatial spillover effect further amplifies this multiplier effect due to the inherent non-equilibrium nature of the financial system. Firstly, the impact of digital financial inclusion on agricultural carbon emissions exhibits significant spatial heterogeneity due to differences in socioeconomic levels among different regions. This is especially evident in the case of relatively economically developed areas with convenient transportation, a large population (sufficient laborers and consumers), and enough fiscal revenue, which is conducive to the accumulation of financial capital.

In contrast, economically backward regions, which may lack sufficient financial capital, benefit from digital inclusion, increasing the possibility of providing financial support to financially backward areas [37]. The marginal benefits from digital inclusion may be more significant in economically underdeveloped regions than in economically or financially developed regions [38]. Secondly, digital financial inclusion has increased the availability of funds in rural areas and the agricultural economy sector. Under the current series of national policies for rural revitalization, digital financial inclusion can reduce the unbalanced allocation of funds and capital in the traditional financial industry. It can also reduce the geographical and sectoral discrimination in finance, providing financial capital for the balanced development of regional agricultural economies and the reasonable distribution of carbon emission intensity among regions. Digital financial inclusion can also reduce information asymmetry and transaction costs, helping rural areas transition to a low-carbon and green development model [39]. With the advantage of big data from digital financial inclusion, it has a spatial spillover impact on regional carbon emission reduction in neighboring areas [40].

Moreover, digital financial inclusion has the dual attributes of "green" and "financing" for agricultural carbon emission reduction, allowing more funds to support rural areas in developing large-scale planting and farming, promoting large-scale renewable energy, and

implementing sustainable agricultural industrialization projects. In addition, funds can be used to support green and low-carbon projects, such as rural water conservancy, surface pollution control, sustainable transport, and other rural infrastructure development, as well as the renovation of low and medium-yield farmland. Diversified funding sources can reduce credit constraints and sources of agricultural carbon emissions, even in relatively economically backward regions. Driven by the above-mentioned "trend towards optimization," digital financial capital will continue concentrating in regions with highly developed green and low-carbon agriculture and higher marginal returns. This will lead to the concentration of financial capital benefits, labor capital benefits, and policy benefits in these regions, accelerating the flow of various resources and capital within the region and creating a particular agglomeration effect of industrial and labor capital in surrounding areas. Based on these observations, this paper proposes the following hypothesis 2:

H2: Digital financial inclusion has solid spatial heterogeneity, influenced by factors such as economic structure, industrial structure, and government policies among regions. It can help reduce the regional gap in agricultural carbon emission intensity.

## 2.3 Mechanism of action

Digital financial inclusion can influence the carbon intensity of industries through technological progress and optimal factor allocation [41]. A well-developed financial market can direct the flow of capital and various financial products to areas with more high-tech projects through market interest rate adjustments, promoting the efficiency of agricultural production in these areas [42]. Additionally, due to multiple carbon emission sources and the fact that agriculture is a significant carbon sink system that often requires government subsidies, this paper believes that obtaining good technical support and sufficient financial capital support is vital to reducing agricultural carbon emissions.

1. The mediating role of technological progress. Technological progress is an intrinsic driver of high-quality regional economic development. In the agricultural sector, technological progress can yield higher marginal returns. Farmers and agriculture are significantly more motivated to adopt new technologies and skills [43]. The government uses traditional fiscal preferential policies to encourage the adoption of new technologies. Also, it relies on financial means to achieve the widespread use of new technologies in rural areas [44]. With the support of digital financial inclusion, the development and application of new technologies can significantly improve the efficiency of energy and resource use in agricultural production. It can expand the boundaries of agriculture towards large-scale and agricultural green output, extend the boundaries of technological innovation in green agricultural production, improve agricultural industrialization and profitability, and promote more regions and enterprises to use green and low-carbon technologies that are conducive to carbon emission reduction. In addition, from the perspective of the decomposition of provincial and industrial targets for achieving China's dual-carbon goals, carbon emissions have become a production cost for the agricultural industry. While a region's agricultural development uses new technologies to improve resource and energy efficiency, it will also consider the cost of carbon emissions. Digital financial inclusion will also consider the cost of reducing carbon in production while supporting relevant technological innovations, thereby increasing the agricultural industry's green and low-carbon technology content. In other words, digital financial inclusion will closely integrate with carbon reduction, technological progress, and guiding more financial resources and services to improve regional agricultural production efficiency and reduce agricultural carbon emissions. Digital financial inclusion can become an "incubator" for developing a green economy in rural areas, accelerating the

transformation of the traditional mode of agricultural industry development and the change of rural economic structure through the accumulation of technological progress and factor inputs. Established studies have also confirmed the technical advancement effect of digital financial inclusion. For example, Suhrab M et al. (2024) found that the technological progress effect of digital financial inclusion on less economically developed regions was more apparent [38]. Digital financial inclusion breaks through the boundaries of land, labor, and other resource supply through technological progress, promoting agricultural resource conservation and inter-regional synergistic reduction of pollution and carbon emissions with mechanized, large-scale, and intensive agricultural development models [42].

2. The mediating role of financial resource allocation capacity. The mediating role of financial resource allocation capacity can better judge the impact channel of local financial support on agricultural carbon reduction. From the perspective of a rational economic agent, the key to promoting agricultural carbon reduction action is to achieve an increase in agricultural GDP and an increase in farmers' income. However, traditional finance may have difficulty providing higher financial support as long as inefficient agricultural development models prevail in rural areas. A certain amount of financial support is needed to achieve these goals. However, it is difficult for traditional finance to do so: firstly, for rural areas, the conventional inefficient agricultural model is still standard, and the agricultural industry is incapable of self-innovation. It is usually tricky for traditional finance to provide higher financial support. Relying on local financial resources or transfer payments is often unsustainable. When a region lacks financial resource allocation capacity for agricultural transformation, the possibility of realizing regional carbon emission reduction targets from an agricultural perspective is weak. Secondly, with the migration of the rural population and the acceleration of urbanization in China, it is difficult to obtain more financial support by relying only on existing low-value collateral in rural areas. The World Bank report (2020) found that promoting sustainable and green agriculture in economically underdeveloped areas requires more robust social financing capacity from local governments. Economically underdeveloped regions and low-carbon agricultural industries with low returns and high upfront investment may find it harder to obtain adequate financial support to reduce carbon emissions from agriculture due to poor access to finance and the propensity of financial institutions to lend. As digital financial inclusion develops, more regions can obtain financial support through digital finance platforms, enhancing local financial support for regional low-carbon and green agricultural development. Specifically, local governments can build and improve digital platforms to strengthen financial allocation capabilities, prompting financial institutions to break through administrative and geographical boundaries and improve the targeting of funds and financial products to the applicants [17, 18]. For local governments with smaller financing capacities, digital financial inclusion platforms provide access to more funding sources. Through digital technology and fintech innovation, the original capital stock can improve the accuracy, speed, and breadth of lending, accelerate the transformation of stock capital, and improve the external environment for financial support of agricultural carbon emission reduction and high-quality development. Digital financial inclusion can also reduce information search costs, mitigate information asymmetries in the structure of agricultural investment among regions, converge the distribution of financial resource returns, and induce a shift of financial resources towards state-supported green and low-carbon agricultural inputs [34]. As a transformative approach to financial services, digital finance has reconfigured local governments' traditional financial resource allocation. Local governments' good financial resource allocation

capacity can provide faster financing channels through digital financial inclusion to help the agricultural industry achieve low energy consumption and high output. In particular, local financial support for low-carbon and green agricultural industries with low returns and long investment cycles enhances the liquidity of funds, provides better resistance to irresistible risks, improves the structure of the agricultural sector, and reduces the gap in the intensity of agricultural carbon emissions among regions. Therefore, this paper proposes the following Hypothesis 3.

H3: Digital financial inclusion can reduce the intensity of agricultural carbon emissions through technological progress and financial resource allocation capacity, narrowing the gap between regional agricultural carbon emission intensities.

## 2.4 Moderating effect

The urban-rural income gap is a significant factor that influences carbon emissions. A higher urban-rural income gap will lead to a migration of the rural population to cities, as various financial elements and services concentrate in cities along with the flow of industry and population. Small-scale farming, decentralized planting, and agricultural development modes in rural areas result in a higher carbon emission intensity [31–33]. Under the promotion of the strategies for common prosperity and rural revitalization, digital financial inclusion can support more intensive, low-carbon, and green agricultural production methods, such as land transfer, large-scale planting, and ecological agriculture. Improving the urban-rural income gap has also released more labor, land, and capital dividends. In addition, digital financial inclusion can encourage farmers to participate in low-carbon and green agriculture by utilizing their own technology, information, and work experience. After obtaining certain benefits, they can continue to expand their operations by increasing the input of funds, technology, and human capital, further improving their income structure. In other words, digital financial inclusion conveys a positive signal of income growth through participation in green and low-carbon agriculture, which helps reduce agriculture's carbon emission intensity [34]. With the continuous extension of digital financial inclusion in rural areas and various fields of the agricultural industry, the convergence of the urban-rural income gap through digital financial inclusion can promote the concentration of human and financial capital in rural areas, resulting in income growth effects. Increasing rural population income can further improve the rural industrial structure and promote industrial structural optimization and transformation by improving resource and energy efficiency. Therefore, the narrowing urban-rural income gap has enabled the development of digital financial inclusion in rural areas, which generates a driving mechanism for regional agricultural production and helps to deliver information to rural areas. This promotes expanding financial resources into key national development areas, such as low-carbon and green agricultural industrialization. It also encourages more areas to choose low-carbon and environmentally friendly agricultural development methods, promotes the use of related green and low-carbon innovative technologies, and improves agricultural productivity, ultimately promoting interregional convergence of agricultural carbon intensity. Based on this, the paper proposes the following Hypothesis 4.

H4: The rural-urban income gap positively moderates the convergence of provincial differences in agricultural carbon emission reduction intensity due to digital inclusion.

All assumptions are shown in Fig 1.

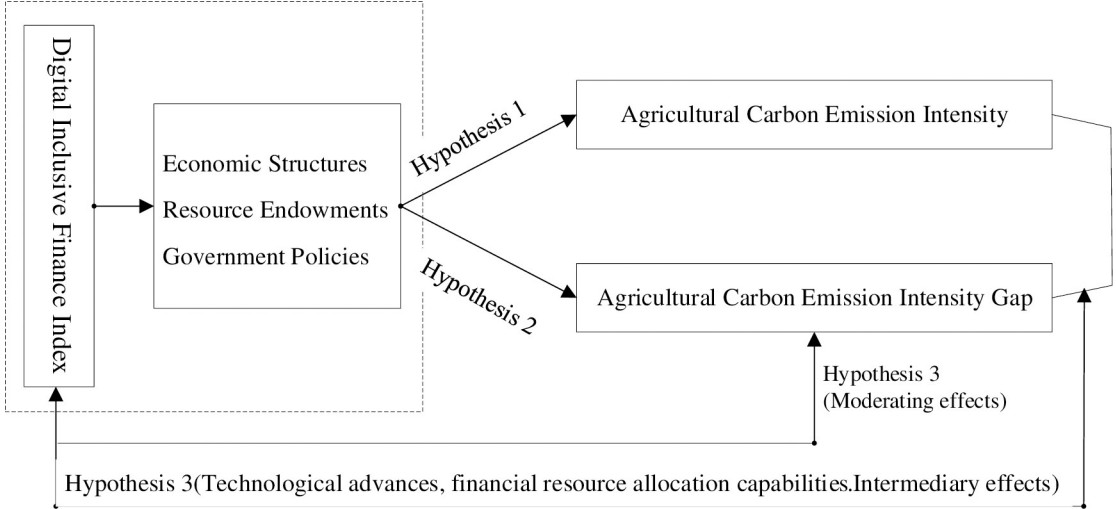

**Fig 1. Schematic diagram of the research hypothesis.**

## 3. Data sources, variable selection, and research methodology

### 3.1 Data sources and variable selection

This article uses the 2010–2020 digital financial inclusion index matched with the same year's provincial agricultural carbon emission intensity as the empirical analysis dataset. Indicators are primarily derived from literature and statistics. They include the following three parts: one part is digital financial inclusion data. This data is mainly from the Digital Financial Inclusion Index *(Dfi)* published by the Digital Finance Research Centre of Peking University, which covers the provincial level [45]. It mainly includes the breadth of coverage (*Wid*), depth of use (*Dep*), and degree of digital services (*Dig*).

The second part is inter-provincial agricultural carbon emission intensity data. Precisely, the provincial narrow agricultural (planting) carbon emission intensity (*Eagri*) and the provincial agricultural carbon emission intensity Degum Gini coefficient (*D-Gini*) were calculated [22]. Calculations were conducted from six aspects: fertilizers, pesticides, agricultural film, diesel, tillage, and agricultural irrigation [43].

The third part is provincial characteristics variables, which mainly include four categories. The first is socio-economic indicators. Different socio-economic factors can affect the total GDP of the regional agricultural sector, primary capital inputs, and other factors, ultimately involving the total agricultural carbon emissions and changes in carbon emission intensity values [46]. They include industrial structure (*IS*), expressed as the proportion of agricultural output value to total agricultural, forestry, animal husbandry, and fishery output value; output value per unit area of major crops (*Vout*), expressed as the ratio of the output value of significant crops to the total area of agricultural land (*Million yuan/ha*); absolute power of agricultural machinery (*Mec*), including the absolute power of all kinds of power machinery used in agriculture, forestry, animal husbandry, and fishery (*Million kilowatts*). The second is government policy support. Agricultural production is mainly a low-yield, market-vulnerable industry compared with the secondary and tertiary industries. Its low-carbon and green development cannot be achieved without the government's public financial support. Therefore, fiscal expenditure is selected as an indicator to measure the government's support for the agricultural industry (*Finan*), explicitly using the ratio of agricultural, forestry, and water conservancy expenditures to GDP to characterize it [47]. The third is ecological and

environmental indicators. The Chinese government has continuously promoted coordinated pollution and carbon reduction implementation. Therefore, reducing agricultural pollution can help lower the agricultural industry's total carbon emissions [48]. They include the discounted amount of agricultural fertilizer application (*Chem*), the total amount of fertilizer used in the study area in one year (*Million tons*), the total amount of agricultural fertilizer used (*Fert*), the total amount of fertilizer used in the study area in one year (*Million tons*). The fourth is human capital security. Higher levels of human capital can better utilize digital financial inclusion and provide the necessary skills for the low-carbon and high-quality development of the agricultural industry, thereby reducing the carbon emission intensity of the industry. The average years of education of the rural population were selected to characterize human capital (*Edu*), Edu = 0 for illiterate/semi-literate, 6 for primary school, 9 for middle school, 12 for high school, and 16–22 for college and above [43].

The fourth part is the mediating variable, which includes technological progress (*Tech*). The ratio of R&D expenditure to the total GDP in the research area in the current year is used to measure this. In contrast, the financial resource allocation ability (*Cred*) is calculated using the ratio of regional GDP to the social financing scale [43].

The fifth part is the moderating variables, where the urban-rural income gap is mainly used for description. It is precisely calculated using the Thayer index (*Theil*).

Due to the lagging nature of the impact of digital finance [3], carbon emission reduction is more susceptible to historical emissions, and there is an extreme path dependence. Therefore, digital finance indicators, carbon emission indicators, and some macroeconomic indicators are analyzed with a one-period lag. Trend values are used to obtain some of the data. Also, considering the zero-income case, a 5% upper and lower tail of income is entered into the moderation effect model analysis. In addition, the logarithmic treatment of the data does not change the original characteristics of the data and reduces the estimation bias by reducing heteroskedasticity. Therefore, some data are entered into the model analysis in logarithmic form and deflated using 2000 as the base year. The macro data above is mainly from the China Statistical Yearbook.

## 3.2 Model setting

The general measurement method cannot be easily extended to non-central locations, and only observes the average treatment effect of digital inclusion, ignoring the impact of digital inclusion on regional heterogeneity of interprovincial agricultural carbon emission intensity, such as the impact of digital inclusion on high agricultural carbon emitting provinces (the upper tail) versus low agricultural carbon emitting provinces (the lower tail), which are often the two groups that have the most direct role in the effect of the carbon agricultural emission reduction effect of digital inclusion Firpo et al. (2018) further expand on the technique of conditional quantile regression with the proposed regression of the regrouping influence function (RIF) [49]. which assesses the effect of changes in the distribution of the explanatory variables on the quantiles of the unconditional (marginal) distribution of the outcome variable, and which achieves an essential breakthrough from focusing on the decomposition of differences in the means to the decomposition of differences in the entire distribution, and is capable of give the degree of influence of each independent variable, better avoiding the endogeneity problem arising from omitted variables. Based on this, to test hypotheses H1 and H2, we used the regression of the reaggregated impact function (RIF) method to examine the impact of digital financial inclusion on inter-provincial changes in agricultural carbon intensity and variation in China. The RIF method avoids endogeneity problems arising from omitted variables [49]. It is commonly used in measures of income disparity and wealth gap. The specific

procedures involved three steps: First, we conducted RIF regression analysis using the conditional mean function, quantile points, and Dagum Gini coefficient to test whether digital financial inclusion affects inter-provincial changes in agricultural carbon emission intensity. Second, we used a mediation effect model to analyze the mechanism of the role of digital financial inclusion on the impact of regional disparities in the intensity of agricultural carbon emission reductions. Third, we applied the RIF-Blinder-Oaxaca model, which constructs a counterfactual distribution function to decompose the factors influencing the interprovincial gap in agricultural carbon emission reduction intensity. Finally, we used a moderating effect model to verify the impact of the moderating variables.

### 3.2.1 Baseline model

According to the research objectives and previous literature, we constructed the following RIF regression model:

$$RIF\{lny_{it}, v^{D-G}(F_Y)\} = \beta_0 + \beta_1 X_{it-1} + \beta_2 Z_{it-1} + \emptyset + \eta + \varepsilon_{it} \tag{1}$$

$$RIF(lny_{it}; q_r, F) = q_r + \frac{\tau - I(lny_{it-1} \leq q_r)}{f_{lny}(q_r)} \tag{2}$$

$$RIF(lny_{it}|\emptyset; \gamma; q_r) = B\tilde{X}\tilde{Z}|\emptyset + \varepsilon_{it} \tag{3}$$

To verify the establishment of H1, the base regression model adopts the regression method of regrouping influence function (*RIF*), as shown in Eq (1); to consider the changes in the intensity of interprovincial agricultural carbon emission reductions at different quartiles, an unconditional quartile regression model is constructed based on the decentralized mapping function shown in Eq (2), as shown in Eq (3). In Eqs (1)–(3), $lny_{it}$ denotes the value of agricultural carbon intensity in region i in year t, and $lny_{it-1}$ is the value of lagged one-period carbon intensity. The term $v^{D-G}(F_Y)$ indicates the Dagum Gini coefficient of $lny$, which reflects the degree of imbalance in carbon intensity in province i in year t. X represents the digital financial inclusion index, further divided into the breadth of coverage, depth of use, and degree of digital services. $X_{it-1}$ is the lagged one-period index value for the sample provinces. $\beta_0$ is the intercept term, $\beta_1$ and $\beta_2$ are the regression coefficient and $Z_{it-1}$ is a set of macro control variables lagged by one period i denotes the sample province, and t represents the year. $\emptyset$ signifies province and industry fixed effects, η refers to time-fixed effects, and $\varepsilon_{it}$ is a random disturbance term. τ represents quantile points, $q_r$ is the unconditional quantile function of $lny$, $f_{lny}(q_r)$ is the probability density function, and B is the vector matrix of regression coefficients. $\tilde{X}$ and $\tilde{Z}$ are the vector matrices of the core explanatory and control variables.

Referring to the 2006 National Greenhouse Gas Inventory Guidelines Carbon Emission Factor Methodology recommended by the IPCC: Calculation of carbon intensity in agriculture

$$TC = \sum C_i = \sum S_i \times \rho_i \tag{4}$$

Where TC is the total agricultural carbon emissions, $C_i$ denotes the total agricultural carbon emissions of the ith carbon emission source, $S_i$ refers to the total carbon emission input of the ith carbon emission source and $\rho_i$ represents the emission factor of each carbon emission source. The reference coefficients and literature sources used in the equation are presented in Table 1.

**Table 1. Carbon emission factors for primary carbon sources.**

| Carbon source factor | Carbon emission factor | Literature sources |
|---|---|---|
| Fertilizer | 0.896 kgC.kg$^{-1}$ | Oak Ridge National Laboratory |
| Pesticides | 4.934 kgC.kg$^{-1}$ | Oak Ridge National Laboratory |
| Agricultural film | 5.180 kgC.kg$^{-1}$ | The Institute of Resource, Ecosystem, and Environment of Agriculture (IREEA) of Nanjing Agricultural University |
| Diesel | 0.593 kgC.kg$^{-1}$ | Intergovernmental Panel on Climate Change |
| Ploughing | 312.603 kgC.kg$^{-1}$ | Wu FL et al. (2008) [50] |
| Irrigation | 20.476 kgC.kg$^{-1}$ | Duan HP et al. (2012) [51] |

Calculate the value of agricultural carbon intensity according to Eq (4):

$$E_{agri} = TC_{it}|GDP_{it} \tag{5}$$

In the equation, $E_{agri}$ represents the value of agricultural carbon emission intensity (KG/10000). GDP refers to the total agricultural output value of the sample provinces, and both variables are included in the equation using logarithmic values.

In addition, the Dagum method was used to measure the differences in agricultural carbon emission intensity between regions in China. At this stage, the indicators for studying provincial differences mainly include the coefficient of variation, Mean Log Deviation Index ($MLD$), Tyrell's index Gini coefficient, etc. However, when analyzing the above methods, most of them cannot decompose the provincial differences or do not consider the distribution of sample subgroups when deteriorating, so they all have certain defects in the application. Some researchers have used the Dagum Gini coefficient to overcome the above difficulties to analyze provincial dynamic differences and evolutionary trends. In this paper, the Dagum Gini coefficient is utilized to measure the inter-provincial differences in the intensity of agricultural carbon emissions in China and decompose them to obtain the composition of the relative differences between inter-provincial regions.

$$G = \sum_{j=1}^{k} \sum_{h=1}^{k} \sum_{i=1}^{C_j} \sum_{r=1}^{C_h} |y_{ji} - y_{hr}|/2C^2\overline{y} \tag{6}$$

In the equation, $y_{ji}$ ($y_{hr}$) denotes the agricultural carbon intensity of any provincial unit within the $j(h)$ region, C is the number of sample provinces, is the national average of agricultural carbon intensity, k is the number of regional divisions, and $C_j(C_h)$ is the number of provincial units within the $j(h)$ region.

$$G_{jh} = \sum_{i=1}^{G_j} \sum_{r=1}^{G_h} |y_{ji} - y_{hr}|/C_j C_h (\overline{Y}_j + \overline{Y}_h) \tag{7}$$

Eq (7) is the Dagum Gini coefficient of agricultural carbon intensity between provinces j and h. $\overline{Y}_j$ and $\overline{Y}_h$ are the average agricultural carbon emission values of provinces j and h.

**3.2.2 Mediating effects model.** To test hypothesis H3, a mediating effects model was used to test the mediating effects of technological progress, the government's ability to allocate financial resources, and the mediating effects of the combination of the two, respectively.

$$RIF\{M_n; v^{DG}(F_Y)\} = \gamma_0 + \gamma_n X + \lambda_2 Z + \emptyset + \mu \tag{8}$$

$$RIF\{ln_{y_{it}}, v^{DG}(F_Y)\} = k_0 + \sum_{n=1}^{2} \varphi_n M_n + k_1 X_{it} + k_2 Z_{it-1} + \emptyset + \delta \tag{9}$$

$$\beta_1 = k_1 + \sum\nolimits_{n=1}^{2} o_n \gamma_n \tag{10}$$

In the equation, μ and δ are random disturbance terms, $M_n$ the mediating variable includes technological progress and financial resource allocation capacity. $\gamma_0$ and $k_0$ are intercept terms. $\gamma_n$ and $\lambda_2$ are regression coefficients, $k_1$ denotes the direct effect of the core explanatory variables, and $\beta_1$ is the total effect of the impact of digital financial inclusion on the intensity of carbon reduction in agriculture and the gap in the intensity of carbon reduction in agriculture. $\varphi_n$ and $\gamma_n$ refer to individual mediating effects, and $\beta_1 - k_1$ signifies the joint mediating effect. If $\gamma_n$ in Eq (8) and $\varphi_n$ in Eq (9) are both significant, it can be shown that digital financial inclusion can reduce agricultural carbon emission intensity and Dagum Gini coefficient through the mediating variable $M_{it}$. Conversely, if they are not significant, the opposite is true.

**3.2.3 Moderating effects model.** Considering the moderating effect of urban-rural income disparity on the convergence of regional agricultural carbon emission reduction intensity, the following regression model of moderating variables was set [52].

$$RIF[lny_{it}; v^{DG}(F_Y)] = a_0 + a_1 X_{it-1} + a_2 T_{it} + a_3 X_{it-1} \times T_{it} + a_4 Z_{it-1} + \emptyset + \epsilon_{it} \tag{11}$$

In Eq (11), $a_0$ is the intercept term, $a_1$-$a_4$ are the regression coefficients and $\epsilon_{it}$ is the random disturbance term. T is the moderating variable. $X_{it-1} \times T_{it}$ is the interaction term between the digitalinclusive finance index and the urban-rural income gap. The interaction variables are also decentered to avoid possible co-linearity between the variables.

$$RIF[lny_{it}; v^{DG}(F_Y)] = a_0 + a_1 X_{it} + a_2 T_{it} + a_3 (X_{it-1} - \overline{X_{it-1}})(T_{it} - \overline{T_{it}}) + a_4 Z_{it-1} + \emptyset + \epsilon_{it} \tag{12}$$

Suppose the coefficient of the interaction term is significantly positive and the model fit is better in model (12) compared to model (11). In that case, it can be judged that the urban-rural income gap can strengthen the convergence effect of digital financial inclusion on provincial differences in agricultural carbon emission intensity.

In addition, the Thiel index was used to describe the urban-rural income gap to avoid data errors arising from differences in statistical caliber. The specific formula is as follows:

$$Theil = \sum\nolimits_{j-1}^{2} \left(\frac{y_{jt}}{y_t}\right) \times ln\left[\left(\frac{y_{jt}}{y_t}\right) / \left(\frac{x_{jt}}{x_t}\right)\right] \tag{13}$$

In Eq (13), j represents urban (j = 1) or rural (j = 2). $y_{jt}$ represents urban or rural disposable income in year t, and $y_t$ represents overall urban and rural disposable income in year t. $x_{jt}$ represents the total urban or rural population in year t, and $x_t$ is the total urban and rural population in the sample provinces in year t.

**3.2.4 Decomposition of the regional agricultural carbon intensity gap.** The RIF-Blinder-Oaxaca decomposition method was used to find the main factors affecting agriculture's regional carbon emission reduction intensity gap. Specifically, the total difference in regional carbon emission reduction intensity was decomposed into the structural effect of the difference in regional agricultural carbon emission reduction intensity caused by digital financial inclusion between different regions and the characteristic impact of regional agricultural carbon emission intensity differences triggered by differences in socio-economic, resource endowment and other characteristic values between different regions.

The regional agricultural carbon emission intensity gap decomposition method combines the implications of the regression of the regrouping influence function (*RIF*) method, which decomposes the total interprovincial agricultural carbon emission intensity differences into

structural and characterization effects by constructing a counterfactual distribution function, to obtain the specific contribution of each explanatory variable to the total differences, and extends it to RIF regressions of the Gini coefficient and the Atkinson's index. Among them, the structural effect is determined by the difference in total agricultural carbon emissions between different inter-provinces caused by the additional shares of agricultural GDP; the characterization effect is caused by the differences in the individual characteristics of these two groups, such as the basis of socio-economic development and differences in human capital.

### 3.3 Descriptive statistical analysis

Table 2 presents the results of descriptive statistics. The sample of 30 provinces has a mean value of 2.273, which is in the higher quartile range. This indicates that a considerable number of provinces have higher agricultural carbon emission intensity. The mean value of the Dagum Gini coefficient of agricultural carbon emission intensity between provinces is 0.611, with a significant difference between the maximum and minimum values, indicating that the difference in agricultural carbon emission intensity between provinces is gradually increasing. The mean value of the digital financial inclusion index is 2.283. Among the mean values of the three dimensions of digital financial inclusion (breadth of coverage, depth of use, and level of digitalization), the level of digitalization has the highest value. However, the mean value of the breadth of coverage is relatively small, and the variance of all three dimensions is increased.

The results of the descriptive statistics of the control variables show that the mean value of the industrial structure is 0.474, with a standard deviation of 0.093. The value of agricultural output accounts for 47% of the total output value of agriculture, forestry, animal husbandry, and fishery. The mean output value per unit area of major crops is 5.179, with a standard deviation of 1.767, indicating a slight difference in the output value per unit area of major crops. The same applies to agricultural machinery power. The mean value of agricultural financial

**Table 2. Variables and descriptive statistics.**

| Variable type | Variable names and symbols | Mean value | Standard deviation | Maximum value | Minimum value |
|---|---|---|---|---|---|
| Explained variables | Agricultural carbon emission intensity (*Eagri*) | 2.273 | 0.262 | 0.793 | 2.764 |
| | Dagum Gini coefficient for agricultural carbon intensity at the provincial level (*D-Gini*) | 0.611 | 0.496 | 0.649 | 0.123 |
| Explanatory variables | Digital financial inclusion Index (*Dfi*) | 2.283 | 1.732 | 2.087 | 2.532 |
| | The breadth of coverage (*Wid*) | 2.231 | 1.767 | 1.762 | 2.506 |
| | Depth of use *(Dep)* | 2.307 | 1.771 | 1.954 | 2.601 |
| | Level of digitalization (*Dig*) | 2.418 | 2.084 | 2.341 | 2.643 |
| Control variables | Industrial structure (*IS*) | 0.474 | 0.093 | 0.767 | 0.223 |
| | Value of output per unit area of major crops (*Vout*) | 5.179 | 1.767 | 5.767 | 4.972 |
| | Agricultural machinery power (*Mec*) | 5.543 | 2.275 | 5.967 | 4.964 |
| | Agricultural financial expenditure (*Finan*) | 0.156 | 0.108 | 0.059 | 0.674 |
| | Fertilizer application for agricultural use in discounted amounts (*Fert*) | 3.768 | 0.929 | 3.779 | 3.721 |
| | Total agricultural pesticide use (*Chem*) | 2.246 | 0.652 | 2.257 | 2.144 |
| | Human capital (*Edu*) | 7.623 | 2.749 | 0 | 22 |
| Intermediate variables | Technological progress (*Tech*) | 0.023 | 0.029 | 0.000 | 0.426 |
| | Financial resource allocation capacity (*Cred*) | 0.317 | 0.064 | 0.009 | 0.934 |
| Adjustment variables | Urban-rural income gap (*Theil*) | 0.423 | 0.091 | 0.184 | 0.701 |

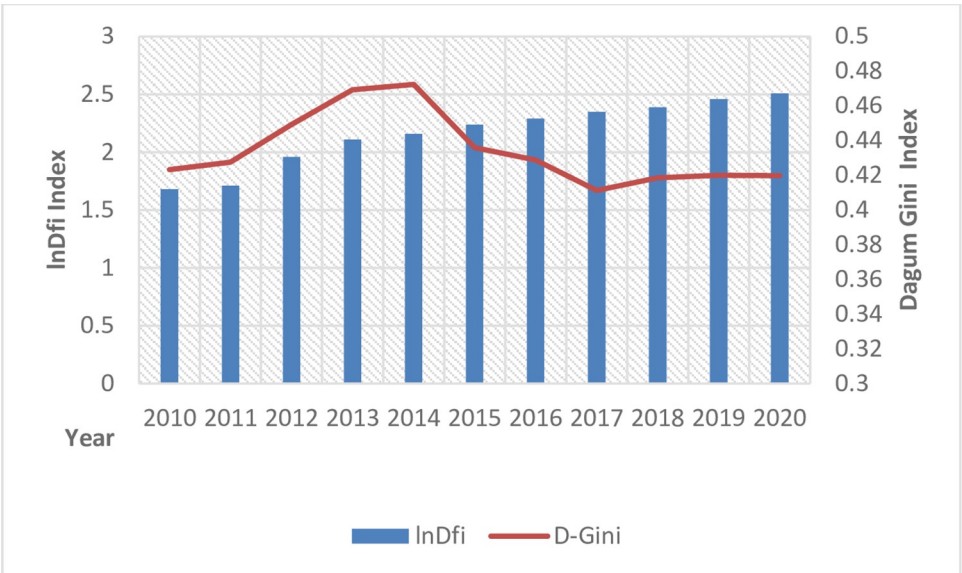

**Fig 2. Trends in Dagum Gini coefficients for digital financial inclusion and agricultural carbon intensity.**

expenditure is low, and the difference between the maximum and minimum values is significant, indicating that agricultural financial spending is lacking in most areas. The standard deviation of pesticide and chemical fertilizer use shows that the total amount used varies significantly between regions. There is a significant difference between the maximum values of educational attainment among individuals, with the average number of years of education being around 7.6, which means that most people have not completed nine years of compulsory schooling. Among the intermediate variables, technological progress and financial resource allocation capacity have small mean values and a large difference between the maximum values. As for the moderating variables, the urban-rural income gap Thiel index is 0.423, and although the urban-rural income gap in China has converged, there is still a large gap between provinces.

As shown in Fig 2, the Dagum Gini coefficient remained above the average from 2010 to 2013, reached its maximum value in 2014, and started declining. However, it showed a flat decline throughout the study period, opposite to the digital financial inclusion index trend. Fig 3 shows the top ten provinces ranked by the Dagum Gini coefficient of agricultural carbon emission intensity. Provinces such as Anhui, Hunan, Hubei, Yunnan, and Jiangxi consistently occupied the top five positions in the Dagum Gini coefficient. These provinces often faced issues, such as extensive agricultural development and frequent grain cultivation activities. To some extent, this reflects that while the share of agricultural GDP in China decreased, agricultural production was gradually concentrated in some provinces and cities, leading to higher carbon emissions in some areas and an increase in the regional carbon emission intensity Dagum Gini coefficient.

# 4. Analysis of empirical results

## 4.1 Analysis of baseline model regression results

Before conducting the regression analysis, we used a mixed, random each models The BP and Hausman tests indicated that the fixed-effects model was more appropriate. Given the potentially large differences in carbon intensity between sectors, the model also controlled for sectoral fixed effects.

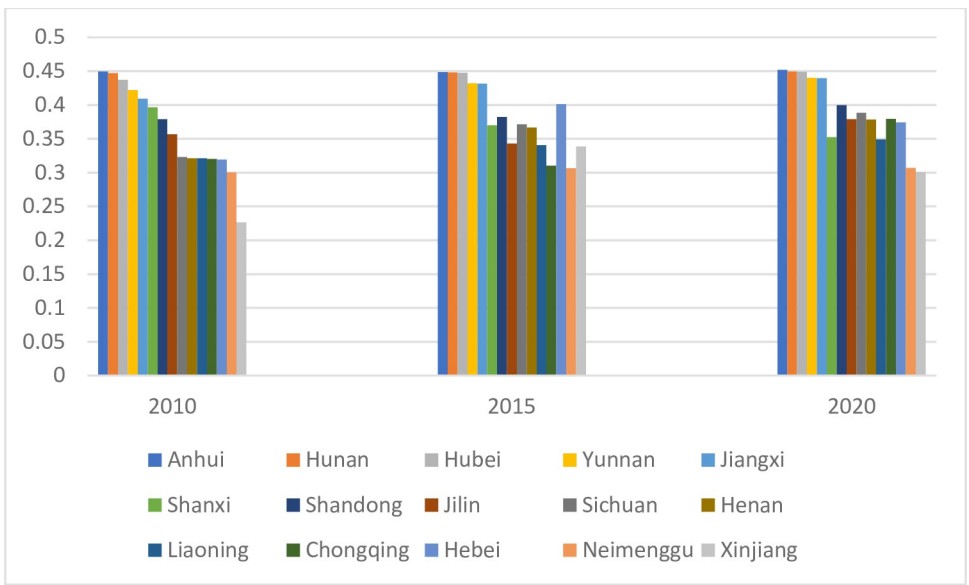

**Fig 3. Trends in inter-provincial agricultural carbon emission Dagum Gini coefficient rankings, 2010–2020.**

Table 3 presents the results of the impact of digital financial inclusion on China's agricultural carbon emission intensity and the regional agricultural carbon emission intensity gap. Column (1) shows the conditional mean model without control variables, while columns (2) and (3) show the regression results after adding control variables. Column (3) displays the regression results of the regional agricultural carbon emission intensity Dagum Gini coefficient. As shown in Table 3, digital financial inclusion has a significant negative effect at the 1% statistical level. For every 1% increase in the digital financial inclusion index, there is a 0.329% decrease in the value of carbon emission intensity and a 0.096% decrease in the Dagum Gini coefficient. This suggests that digital financial inclusion can significantly reduce the value of agricultural carbon emission intensity and narrow the regional agricultural emission intensity gap. Therefore, research hypotheses H1 and H2 are partially supported.

The results of the analysis of control variables show that the industrial structure has a negative and significant effect on the Dagum Gini coefficient of agricultural and regional carbon emission intensity. When the output value per unit area of significant crops increases, the value of agricultural carbon emission intensity and the inter-regional agricultural carbon emission intensity Dagum Gini coefficient decreases, for each 1% increase in agricultural mechanization power, there is a 0.017% decrease in the regional agricultural carbon emission reduction intensity Dagum Gini coefficient. Suppose the government invests more financial resources in agricultural development projects. In that case, it will reduce the value of agricultural carbon emission intensity and narrow the gap in carbon emission intensity between regions. The increase in the use of pesticides and fertilizers increases the value of agricultural carbon intensity to varying degrees and widens the carbon intensity gap between regions. Individuals with longer education years tend to have higher education levels, making them more receptive to low-carbon and green industrial development models and technologies. This enables them to adopt these models and technologies in production, reducing agriculture's carbon emission intensity. Additionally, it promotes the convergence of the regional agricultural carbon emission intensity gap.

Columns (1)-(9) of Table 4 present the results of the three-dimensional indices of digital financial inclusion on the regional agricultural carbon emission intensity and the intensity of

**Table 3. Results of the baseline regression analysis.**

| Variables | Conditional Means | Conditional Means | Dagum Gini coefficient |
|---|---|---|---|
| | (1) | (2) | (3) |
| lnDfi | -0.447*** | -0.329*** | -0.096*** |
| | (0.389) | (0.712) | (0.833) |
| IS | | 0.247*** | 0.064** |
| | | (0.083) | (0.121) |
| Vout | | -0.412*** | -0.071*** |
| | | (0.117) | (0.629) |
| Mec | | -0.328*** | -0.017*** |
| | | (0.103) | (0.309) |
| Finan | | -0.261*** | -0.035*** |
| | | (0.338) | (0.737) |
| Fert | | 0.675*** | 0.052*** |
| | | (0.235) | (0.174) |
| Chem | | 0.258*** | 0.029*** |
| | | (0.086) | (0.013) |
| Edu | | -0.137*** | -0.032*** |
| | | (0.114) | (0.043) |
| Provincial fixed effects | Control | Control | Control |
| Time fixed effects | Control | Control | Control |
| Industry fixed effects | Control | Control | Control |
| RIF mean value | 6.743 | 6.629 | 6.784 |
| $R^2$ | 0.169 | 0.096 | 0.232 |
| N | 30 | 30 | 30 |

Notes

a. Standard deviation in brackets.

b. ***, **and * represent significant at the 1%, 5% and 10% levels respectively.

the Dagum Gini coefficient. The difference between them is that columns (1) (4) (7) are the conditional mean model without control variables, while columns (2) (5) and (8) are the regression results after adding the control variables. As shown in Table 4, the three indices of digital financial inclusion can significantly reduce the value of agricultural carbon emission intensity and the regional Dagum Gini coefficient. For every 1% increase in the breadth of coverage, the value of agricultural carbon intensity and Dagum Gini decrease by 0.199% and 0.021%, respectively. For every 1% increase in depth of use, the agricultural carbon intensity value and Dagum Gini decrease by 0.123% and 0.012%, respectively. For each 1% increase in the digital service index, the value of agricultural carbon intensity and Dagum Gini decrease by 0.109% and 0.007%, respectively. The most significant impact is from the breadth of coverage, which indicates that the more comprehensive the coverage of digital financial inclusion, the more effective it is in reducing carbon emission intensity and the Dagum Gini coefficient of agriculture. The more deeply the two are integrated, the more rapid the decline of the two affected indicators in various aspects.

## 4.2 The unconditional quantile regression model

As shown in Table 4, digital financial inclusion can decrease the regional agricultural carbon emission intensity and narrow the gap in the carbon emission intensity within the region. However, the change in the Dagum Gini coefficient implies that allocating digital financial

**Table 4. Impact of the three-dimensional Indices of digital financial inclusion on the regional agricultural carbon emissions gap.**

| Variables | Conditional Means(1) | Conditional Means(2) | Dagum Gini coefficient (3) | Conditional Means(4) | Conditional Means(5) | Dagum Gini coefficient(6) | Conditional Means(7) | Conditional Means(8) | Dagum Gini coefficient(9) |
|---|---|---|---|---|---|---|---|---|---|
| lnWid | -0.234*** | -0.199** | -0.021*** | | | | | | |
| | (0.049) | (0.339) | (0.343) | | | | | | |
| lnDep | | | | -0.203*** | -0.123*** | -0.012*** | | | |
| | | | | (0.504) | (0.273) | (0.364) | | | |
| lnDig | | | | | | | -0.174*** | -0.109*** | -0.007*** |
| | | | | | | | (0.035) | (0.512) | (0.293) |
| Control variables | Control | Control | Control | Control | Control | Control | Control | Control | Control |
| Provincial fixed effects | Control | Control | Control | Control | Control | Control | Control | Control | Control |
| Time fixed effects | Control | Control | Control | Control | Control | Control | Control | Control | Control |
| Industry fixed effects | Control | Control | Control | Control | Control | Control | Control | Control | Control |
| RIF mean value | 7.021 | 6.993 | 6.275 | 5.439 | 6.236 | 6.921 | 7.156 | 7.942 | 6.047 |
| R2 | 0.203 | 0.087 | 0.075 | 0.152 | 0.219 | 0.275 | 0.189 | 0.094 | 0.147 |
| N | 30 | 30 | 30 | 30 | 30 | 30 | 30 | 30 | 30 |

Notes

a. Standard deviation in brackets.

b. ***, **and * represent significant at the 1%, 5% and 10% levels respectively.

inclusion-related services and resources among provinces must be adjusted based on their actual requirements. Therefore, we used an unconditional quantile regression model to accurately describe the impact of digital financial inclusion on the regional gap in carbon emission reduction intensity. We considered the impact effects under different quantitative points from the perspective of digital financial inclusion and its three dimensions: breadth of coverage, depth of use, and degree of digital services, respectively (Table 5).

According to the unconditional quantile regression coefficients in Table 5, the higher the agricultural carbon intensity quantile, the smaller the regression coefficient of the digital financial inclusion index, but the regression coefficients are not significant at the 10th and 90th quantile points. In other words, digital financial inclusion faces challenges in promoting agricultural emission reduction in provinces with low and high carbon emission intensity. The negative coefficients of the three-dimensional indices decrease with the increase of quantile, indicating that the impact of digital financial inclusion on provinces with low agricultural carbon emission intensity is more significant than those with high agricultural carbon emission intensity. Among the three-dimensional indices, the negative impact of coverage breadth is more pronounced. Fig 4 shows the quantile effect of digital financial inclusion on the Dagum Gini coefficient. The white areas represent insignificant effects, while the black areas represent significant ones, indicating that digital financial inclusion can significantly narrow the gap in regional agricultural carbon emission intensity.

As shown in Fig 4, the impact of the digital financial inclusion index on the change in the Dagum Gini coefficient is only significantly negative at the 20th-80th quantile point. The coverage breadth has a highly negative effect at all quantile points. The depth of use does not affect the 10th-20th and 90th quantile points substantially, indicating that the depth of use tends to narrow the gap in regional agricultural carbon emission intensity at the middle and upper

**Table 5. Unconditional quantile regression results.**

| Variable | Agricultural Carbon Emission Intensity (*Eagri*) | | | | | | | | |
|---|---|---|---|---|---|---|---|---|---|
| | 10% | 20% | 30% | 40% | 50% | 60% | 70% | 80% | 90% |
| | (1) | (2) | (3) | (4) | (5) | (6) | (7) | (8) | (9) |
| *lnDfi* | -0.061(0.023) | -0.054*** | -0.057*** | -0.043*** | -0.037*** | -0.028*** | -0.024*** | -0.021*** | -0.009 |
| | | (0.008) | (0.015) | (0.022) | (0.011) | (0.084) | (0.027) | (0.014) | (0.056) |
| *lnWid* | -0.078***(0.145) | -0.075*** | -0.071*** | -0.067*** | -0.061*** | -0.058*** | -0.052*** | -0.043*** | -0.037*** |
| | | (0.027) | (0.349) | (0.054) | (0.092) | (0.006) | (0.034) | (0.675) | (0.023) |
| *lnDep* | -0.083(0.358) | -0.092 | -0.034*** | -0.029*** | -0.026*** | -0.022*** | -0.018*** | -0.015*** | -0.023 |
| | | (0.024) | (0.042) | (0.037) | (0.214) | (0.197) | (0.036) | (0.085) | (0.056) |
| *lnDig* | -0.052***(0.113) | -0.049*** | -0.047*** | -0.045*** | -0.031*** | -0.027*** | -0.061 | -0.055 | -0.046 |
| | | (0.034) | (0.034) | (0.017) | (0.062) | (0.424) | (0.071) | (0.543) | (0.409) |
| Control variables | Control | Control | Control | Control | Control | Control | Control | Control | Control |
| Provincial fixed effects | Control | Control | Control | Control | Control | Control | Control | Control | Control |
| Time fixed effects | Control | Control | Control | Control | Control | Control | Control | Control | Control |
| Industry fixed effects | Control | Control | Control | Control | Control | Control | Control | Control | Control |
| RIF mean value | 8.352 | 7.974 | 7.521 | 8.707 | 8.633 | 6.992 | 7.067 | 8.192 | 7.775 |
| R2 | 0.134 | 0.264 | 0.084 | 0.436 | 0.229 | 0.538 | 0.147 | 0.072 | 0.604 |
| N | 30 | 30 | 30 | 30 | 30 | 30 | 30 | 30 | 30 |

Notes

a. Standard deviation in brackets.

b. ***, **and * represent significant at the 1%, 5% and 10% levels respectively.

quantiles. The lower gap provinces are mainly developed regions of secondary and tertiary production, where the degree of agricultural intensification is usually higher, and coupled with more stringent carbon emission constraints, their agricultural development itself has a strong incentive to reduce carbon emissions and digital financial inclusion tends to lack the incentive to develop deeper. In regions with high gaps, agricultural production tends to be characterized by dispersed cultivation patterns, and the low commercialization and industrialization of agriculture make it challenging to promote the deepening of digital financial inclusion. The degree of digitization only plays a role at the 10th-60th quantile points, indicating that the degree of digitization can effectively reduce the gap in agricultural carbon emission intensity in regions with a moderate quantile.

Table 6 further illustrates the impact of digital financial inclusion for each sample province at different Dagum Gini coefficient quantiles. As shown in Table 6, the Dagum Gini index can significantly affect the convergence of the agricultural carbon emission intensity gap in some provinces, mainly concentrated at the 20th-80th quantile points. However, eastern provinces, such as Beijing, Shanghai, Guangdong, Jiangsu, and Zhejiang, and western provinces, such as Guizhou and Qinghai, were not influenced. Digital services require users to have high financial knowledge and digital literacy. This enables them to efficiently utilize data technology and financial services provided by digital finance platforms to promote low-carbon transformation in the agricultural industry. As a result, provinces with a higher proportion of agricultural population and lower average education levels are more likely to experience a "digital divide," making it difficult for digital financial inclusion to have a significant impact. The breadth of coverage better reflects the inclusiveness and universality of digital financial inclusion, thus influencing all sample provinces. The profound effect is mainly concentrated in regions with well-developed agricultural industrialization or commodity grain planting bases, where the level of digitalization services has reduced the gap in agricultural carbon emission intensity

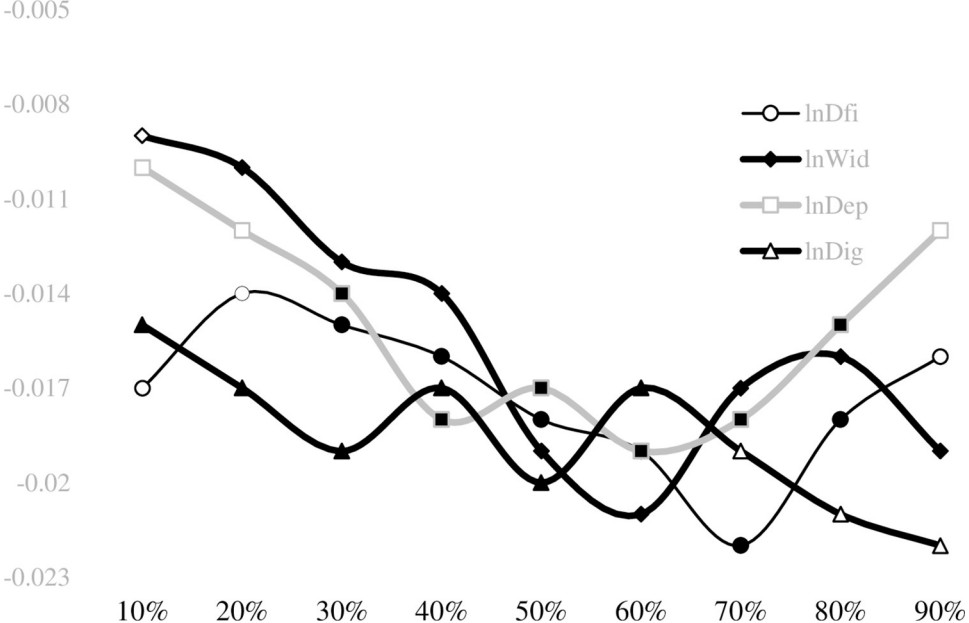

**Fig 4. The plot of the quantile effect of China's digital financial inclusion on the Dagum Gini coefficient of agricultural carbon intensity.**

between economically developed regions and highly developed regions with agricultural industrialization.

## 4.3 Mechanism analysis

Based on the previous theoretical analysis, we applied the mediating effect models (6)-(8) to analyze how digital financial inclusion affects the regional agricultural carbon emission reduction intensity gap through both technological progress and the government's financial resource allocation capacity (Tables 7 and 8). Columns (1) and (3) of Table 7 show the effects of digital inclusion on the mediating variables. After controlling for other variables, the

**Table 6. Distribution of digital financial inclusion impact on carbon intensity gap in agriculture by province.**

| | Direction of Influence | Dagum Gini quartiles | Significant impact provinces | Dagum Gini quartiles | No significant effect provinces |
|---|---|---|---|---|---|
| *ln*Dfi | — | 20%-80% | Tianjin, Hebei, Liaoning, Fujian, Shandong, Hainan, Shanxi, Heilongjiang, Anhui, Jiangxi, Henan, Hubei, Hunan, Jilin, Inner Mongolia, Chongqing, Sichuan, Yunnan, Shaanxi, Xinjiang | 10%, 90% | Beijing, Shanghai, Guangdong, Jiangsu, Zhejiang, Qinghai, Guizhou, Gansu, Guangxi, Ningxia |
| *ln*Wid | — | 10%-90% | Hebei, Liaoning, Fujian, Shandong, Hainan, Shanxi, Heilongjiang, Anhui, Jiangxi, Henan, Hubei, Hunan, Jilin, Inner Mongolia, Guangxi, Chongqing, Sichuan, Guizhou, Yunnan, Shaanxi, Gansu, Qinghai, Ningxia, Xinjiang, Beijing, Shanghai, Guangdong, Jiangsu, Zhejiang, Tianjin | | / |
| *ln*Dep | — | 30%-80% | Jiangsu, Zhejiang, Guangdong, Hebei, Liaoning, Fujian, Shandong, Hainan, Heilongjiang, Anhui, Jiangxi, Henan, Hubei, Hunan, Jilin, Inner Mongolia, Chongqing, Sichuan, Yunnan, Shaanxi, Gansu, Xinjiang | 10%-20% | Beijing, Shanghai, Tianjin, Qinghai, Gansu, Guizhou, Guangxi, Ningxia, Shanxi |
| | | | | 90% | |
| *ln*Dig | — | 10%-60% | Jiangsu, Zhejiang, Beijing, Guangdong, Shanghai, Liaoning, Fujian, Shandong, Heilongjiang, Xinjiang, Jiangxi, Henan, Hubei, Hunan, Jilin, Inner Mongolia, Yunnan, Chongqing, Sichuan, Shaanxi | 70%-90% | Guangxi, Gansu, Qinghai, Hebei, Ningxia, Guizhou, Anhui, Shanxi, Hainan |

**Table 7. Results of the test for mediating effect (1).**

| Variables | Tech | | Cred | |
|---|---|---|---|---|
| | Conditional Means(1) | Dagum Gini coefficient (2) | Conditional Means(3) | Dagum Gini coefficient (4) |
| lnDfi | 0.232*** | -0.019*** | 0.127*** | -0.012*** |
| | (0.149) | (0.023) | (0.515) | (0.004) |
| Control variables | Control | Control | Control | Control |
| Provincial fixed effects | Control | Control | Control | Control |
| Time fixed effects | Control | Control | Control | Control |
| Industry fixed effects | Control | Control | Control | Control |
| RIF mean value | 6.467 | 0.235 | 7.018 | 0.333 |
| R2 | 0.087 | 0.121 | 0.192 | 0.091 |
| N | 30 | 30 | 30 | 30 |

Notes

a. Standard deviation in brackets.

b.***, **and * represent significant at the 1%, 5% and 10% levels respectively.

regression results show that provinces with higher digital inclusion indices are more likely to receive financial support for technological investments and enhance local governments' financial resource allocation capabilities. The regression results in columns (2) and (4) show that increasing the digital financial inclusion index can significantly reduce the technology input gap between regions. More local governments have equal access to social financing.

**Table 8. Results of the test for mediating effect (2).**

| | Conditional Means | Dagum Gini coefficient | Conditional Means | Dagum Gini coefficient | Conditional Means | Dagum Gini coefficient |
|---|---|---|---|---|---|---|
| | (1) | (2) | (3) | (4) | (5) | (6) |
| lnDfi | -0.293*** (0.212) | -0.015*** | -0.251*** | -0.014*** | -0.196*** | -0.019*** |
| | | (0.347) | (0.098) | (0.392) | (0.057) | (0.089) |
| Tech | -0.264** | -0.024*** | | | -0.373*** | -0.025*** |
| | (0.392) | (0.215) | | | (0.013) | (0.034) |
| Mediating effect | 0.246 | 0.019 | | | | |
| Cred | | | -0.293*** | -0.022*** | -0.224*** | -0.022*** |
| | | | (0.074) | (0.084) | (0.111) | (0.129) |
| Mediating effect | | | 0.172 | 0.012 | | |
| Joint effects | | | | | -0.227*** | -0.017*** |
| | | | | | (0.126) | (0.026) |
| Joint Mediating effects | | | | | 0.294 | 0.021 |
| Control variables | Control | Control | Control | Control | Control | Control |
| Provincial fixed effects | Control | Control | Control | Control | Control | Control |
| Time fixed effects | Control | Control | Control | Control | Control | Control |
| Industry fixed effects | Control | Control | Control | Control | Control | Control |
| RIF mean value | 8.056 | 0.273 | 8.056 | 0.273 | 8.056 | 0.273 |
| R2 | 0.135 | 0.243 | 0.101 | 0.242 | 0.074 | 0.116 |
| N | 30 | 30 | 30 | 30 | 30 | 30 |

Notes

a. Standard deviation in brackets.

b. ***, **and * represent significant at the 1%, 5% and 10% levels respectively.

Based on Table 7, further regression analysis was conducted based on Eq (10) (Table 8). According to the regression results in columns (1)-(6) of Table 8, the regression coefficients of the conditional means in columns (1), (3), and (5) are unfavorable. The regression coefficients of the mediating variables of the Dagum Gini coefficients in columns (2), (4), and (6) are harmful, and all coefficients are significant. The absolute value of the conditional mean of the Dagum Gini coefficient in Table 8 is 0.293, smaller than the regression coefficient of 0.329 in column (2) of Table 4. The absolute value of the regression coefficient of the Dagum Gini coefficient (0.024) is also smaller than that of 0.096 in column (3) of Table 4. This partially verifies the intermediation effect H3. The absolute values of the mediating effects of the two mediating variables on the Dagum Gini coefficients are 0.019 and 0.012, indicating that technological progress can play a more significant mediating role in reducing the carbon intensity and Dagum Gini coefficients in agriculture. This means that technological progress can play a more significant mediating effect in reducing the carbon reduction intensity and Dagum Gini coefficient of agriculture. Columns (5) and (6) of Table 8 show the joint mediating effect of the two mediating variables. Using Eq (10), we calculate that the absolute values of the combined mediating effects of the two mediating variables in promoting agricultural carbon emission intensity and the Dagum Gini coefficient are 0.294 and 0.021, respectively, indicating that digital financial inclusion can reduce agricultural carbon emission intensity in all aspects and achieve convergence of the regional Gini coefficient through the joint effect of technological progress and the government's financial resource allocation capacity. Higher government financial resource allocation capacity and technological innovation can continuously optimize the regional agricultural industry structure, reduce total carbon emissions, and achieve convergence of the Gini coefficient. This verifies Hypothesis H3.

## 4.4 Moderating effect

The moderating effects model tests the moderating effect of the urban-rural income gap and constructs the interaction term between the Thiel index and digital financial inclusion. According to the regression coefficients in column (3) of Table 9, after controlling for other variables, the Thiel index, which characterizes the urban-rural income gap, is significantly positive; the more significant the urban-rural income gap, the greater the difference in agricultural carbon emission intensity between regions. The regression results of column (4) with the interaction term of the moderating variables show that the interaction term of the digital financial inclusion index and the Thiel index is significantly positive, and the $R^2$ of column (4) increases by 0.002 compared to column (3). These results indicate that the urban-rural income gap significantly moderates digital financial inclusion, and reducing the urban-rural income gap can further narrow the carbon intensity gap between regions. This validates Hypothesis H4.

## 5. Robustness tests

### 5.1 Endogenous discussion

RIF regression and lagged explanatory variables were used to address the endogeneity problem caused by reverse causality, but there were still issues of omitted variables and two-way causality. To address these problems, we used the spherical distance to Hangzhou as an instrumental variable for the analysis. The geographic distance to Hangzhou was measured from each province's capital city to Hangzhou. The specific processing method involved adding 1 to the geographic distance and taking the logarithm as the instrumental variable (Dis), used in a two-stage least squares regression. The first-stage regression results from Table 10 showed a significant correlation between geographic distance and the digital financial inclusion index. The weak instrumental variable and non-identification test results rejected the null hypothesis, confirming the effectiveness of the instrumental variable. The second-stage regression results

**Table 9. Regression results of the moderated effects model.**

| Variables | Dagum Gini coefficient | Theil Index of Urban-Rural Income Disparity | Dagum Gini coefficient | Dagum Gini coefficient |
|---|---|---|---|---|
| | (1) | (2) | (3) | (4) |
| *lnDfi* | -0.079*** | -0.445*** | -0.067*** | -0.041*** |
| | (0.146) | (0.114) | (0.251) | (0.044) |
| *Theil* | | | 0.037*** | 0.021*** |
| | | | (0.015) | (0.081) |
| *lnDfi×Theil* | | | | 0.032*** |
| | | | | (0.014) |
| Intercept term | 1.234*** | 4.063** | 2.082** | 1.969*** |
| | (0.411) | (1.344) | (0.693) | (1.323) |
| Control variables | Control | Control | Control | Control |
| Provincial fixed effects | Control | Control | Control | Control |
| Time fixed effects | Control | Control | Control | Control |
| Industry fixed effects | Control | Control | Control | Control |
| RIF mean value | 6.824 | 0.357 | 6.824 | 6.824 |
| R2 | 0.097 | 0.086 | 0.133 | 0.135 |
| N | 30 | 30 | 30 | 30 |

Notes

a. Standard deviation in brackets.

b. ***, **and * represent significant at the 1%, 5% and 10% levels respectively.

showed that a 1% increase in the digital financial inclusion index resulted in a 0.012% decrease in the Dagum Gini coefficient. This validated the robustness of the baseline regression results.

## 5.2 Robustness tests

Based on the studies by Sun XT et al. (2022) [53] and Gao K et al. (2022) [54], robustness tests were conducted in two ways: First, the explanatory variables were replaced. The digital

**Table 10. Endogeneity estimation results.**

| Variables | Stage 1 | Stage 2 |
|---|---|---|
| | *lnDfi* | Dagum Gini coefficient |
| *Dis* | -0.057*** | |
| | (1.453) | |
| *lnDfi* | | -0.012*** |
| | | (0.175) |
| Control variables | Control | Control |
| Provincial fixed effects | Control | Control |
| Time fixed effects | Control | Control |
| Industry fixed effects | Control | Control |
| N | 30 | 30 |
| $R^2$ | 0.773 | 0.519 |
| Phase I F-value (P-value) | 11673.79 (<0.01) | |
| Kleibergen-Paap rk LM-value (P -value) | 3675.64 (<0.01) | |

Notes

a. Standard deviation in brackets.

b. ***, **and * represent significant at the 1%, 5% and 10% levels respectively.

**Table 11. Robustness test results.**

| Variables | Dagum Gini coefficient | |
|---|---|---|
| | Replacement of core explanatory variables | Replacement measurement model |
| *Dfi*/100 | -0.026*** | |
| | (0.514) | |
| *lnDfi* | | -0.019*** |
| | | (0.306) |
| Control variables | Control | Control |
| Provincial fixed effects | Control | Control |
| Time fixed effects | Control | Control |
| Industry fixed effects | Control | Control |
| adjusted R2 | 0.639 | 0.585 |
| Generalized Likelihood Ratio Test | -9922.664 | -8692.593 |
| N | 30 | 30 |

Notes

a. Standard deviation in brackets.

b. ***, **and * represent significant at the 1%, 5% and 10% levels respectively.

financial inclusion index divided by 100 was used as a proxy variable to analyze its effect on the intensity of agricultural carbon emission reduction and the regional gap in agricultural carbon emission reduction intensity. The regression results in column (1) of Table 11 show that the coefficients were still negative significant, and consistent with the baseline regression results. This indicates that the results were robust.

Second, the econometric model was replaced. Tobit regression analysis was used to test the robustness of the previous results. The estimation results in column (2) of Table 11 indicate that the digital inclusion index remained negative and significant at the 1% statistical level after changing the analysis method. This confirmed that the baseline regression results were robust.

## 5.3 Spatial heterogeneity test

The level of development in digital financial inclusion varies across regions, resulting in significant regional heterogeneity in the intensity of agricultural carbon emissions and the Gini coefficient. Based on the National Bureau of Statistics data, the 30 sample provinces were divided into four regions: East, Central, West, and Northeast. Regional heterogeneity was also examined according to the distribution and climate of China's primary grain production areas using the "Heihe-Tengchong Line" and the classification of grain and non-grain production areas [41]. Building on the baseline regression in Table 4, the interaction terms for different regions were used to analyze the Dagum Gini coefficients of regional agricultural carbon emission intensity. After controlling for other variables, the final regression results are presented in Table 12.

As shown in Table 12, digital financial inclusion significantly impacts the gap in agricultural carbon intensity across regions, and the effect varies considerably among regions. Further seemingly unrelated regression (SUR), chi-square tests, and p-value tests on the disparities within different groups confirm the significant regional heterogeneity in the impact of digital financial inclusion. According to column (1) of Table 12, using the western region as the reference area, the digital financial inclusion index coefficient is 0.025 and significantly negative at

**Table 12. Spatial heterogeneity in the impact of digital financial inclusion on the carbon reduction gap in agricultural regions.**

| Variables | Dagum Gini coefficient | | |
|---|---|---|---|
| | 1 | 2 | 3 |
| | **East, Central, West, Northeastern** | **Heihe-Tengchong Line (Southeast-Northwest)** | **Grain and non-grain-producing regions** |
| *lnDfi* | -0.025* | -0.087 | -0.031* |
| | (0.072) | (0.473) | (0.079) |
| *lnDfi*×East | -0.032*** | | |
| | (0.093) | | |
| *lnDfi*×Central | -0.027*** | | |
| | (0.6049) | | |
| *lnDfi*×Northeastern | - 0.038*** | | |
| | (0.227) | | |
| *lnDfi*×Southeast | | -0.030*** | |
| | | (0.214) | |
| *lnDfi*×Major grain-producing regions | | | -0.047*** |
| | | | (0.012) |
| Provincial fixed effects | Control | Control | Control |
| Time fixed effects | Control | Control | Control |
| Industry fixed effects | Control | Control | Control |
| N | 30 | 30 | 30 |
| $R^2$ | 0.483 | 0.527 | 0.664 |

Notes

a. Standard deviation in brackets.

b. ***, **and * represent significant at the 1%, 5% and 10% levels respectively.

the 10% level. The interaction term coefficients between the digital financial inclusion index and the eastern, central, and northeastern regions are all considerably negative at 1%. Among them, the impact of digital financial inclusion in the northeast region is the most significant, with an absolute value of 0.038, followed by the eastern region, with a total value of 0.032. The northeastern region is a commodity grain-growing base in China, with high agricultural industrialization and scale. Still, the outflow of population and industry has also made it a traditional financial investment area. Digital financial inclusion, with its "low cost, high speed, and wide coverage," can overcome this limitation. Coupled with the gradual reorganization of the country's development strategy for the northeastern region, digital financial inclusion can support the green transformation of agricultural development in the northeast to a greater extent, ultimately promoting the gradual convergence of the gap in agricultural carbon emissions intensity between regions. However, the agricultural carbon intensity gap between regions is still significant and converging slowly under the influence of digital financial inclusion because of the high agricultural carbon emissions intensity in the western region, which is caused by the long-standing small-scale agricultural economic development model and fragile ecological environment, is difficult to change in the short term. Moreover, the insufficient construction, low coverage, and shallow usage depth of digital financial inclusion platforms aggravate the problem. Therefore, it will take a long time for the convergence effect of digital financial inclusion on inter-provincial agricultural carbon emissions intensity in the western region to become evident.

Results for the Southeast-Northwest region bounded by the Heihe-Tengchong Line are shown in column (2). Taking the northwest region as the reference area, the digital financial inclusion index coefficient is 0.087, but not significant. The interaction term coefficient

between the digital financial inclusion index and the southeastern region is -0.03, and the absolute value of the two coefficients is 0.057. This result indicates that compared with the northwest region, digital financial inclusion in the southeastern region can significantly reduce the Dagum Gini coefficient of inter-provincial agricultural carbon emissions intensity. Although the northwest region contains most of China's high agricultural carbon-emitting provinces, digital financial inclusion has not played a significant role in carbon reduction and regional gap convergence. This is closely related to the relatively weak financial market environment, lack of government monetary and economic policy support, and relatively backward agricultural industrialization development capabilities in the northwest region. The coverage, depth of usage, and level of digital technology popularization of digital financial inclusion in the northwest region are much weaker than in the southeastern region.

According to Cheng QW et al. (2022), the total agricultural carbon emissions and intensity in China's thirteen major grain-producing regions are higher than those in the non-grain-producing areas. The regression results in column (3) indicate that a 10% increase in the index leads to a 0.031% decrease in the Dagum Gini coefficient of agricultural carbon emission intensity in grain-producing regions. The coefficient is also negative and significant at the 1% statistical level for non-grain-producing regions. Digital financial inclusion significantly narrows the gap in agricultural carbon emission reduction intensity between grain-producing and non-grain-producing regions, and the effect is more significant in absolute value for the former. This is related to grain cultivation occurring more frequently than other agricultural cultivation during the year, resulting in higher carbon intensity values in grain-producing regions. Therefore, the grain-producing regions need to continue their efforts to reduce the regional carbon intensity gap.

The regression results in Table 12 support hypothesis H2, which states that digital financial inclusion has a significant spatial heterogeneity effect on the agricultural carbon emission intensity gap and can facilitate regional convergence.

## 5.4 Decomposition of interregional disparities

The gap in agricultural carbon intensity between provinces is one of the critical factors affecting the achievement of China's peak carbon neutrality target. A reweighted RIF-OB decomposition model was applied to decompose the gap in agricultural carbon intensity among the eastern, central, western, and northeastern regions (Table 13). The total decomposition results of the eastern and northeastern regions show that the coefficients of the core explanatory variables and intermediate variables of the real, structural, and characteristic effects are all significantly negative at the 1% level.

Digital financial inclusion can more effectively reduce the agricultural carbon emission intensity Dagum Gini coefficient in the eastern and northeastern regions than in the other areas, where the structural effect is the main factor in narrowing the gap between the two regions. Among the intermediate variables, technological progress and financial resource allocation ability significantly contribute to reducing agricultural carbon emission intensity, with technological progress having a more significant impact.

For the eastern-central region, the regression coefficients of the total effect and characteristic effect are significantly negative, while the decomposition coefficient of the structural impact is not significant. Technological progress and government financial resource allocation ability significantly contribute to reducing agricultural carbon emission intensity between the eastern and central regions, with the latter having a more vital significance. This may be because the central region supplies a considerable amount of grain and other agricultural and forestry products in China, making agricultural development transformation more difficult in this

**Table 13. Re-weighted RIF-OB decomposition results.**

**1**

| | East and Northeastern | | East and Central | | East and West | |
|---|---|---|---|---|---|---|
| | Dagum Gini coefficient | Atkinsom Index | Dagum Gini coefficient | Atkinsom Index | Dagum Gini coefficient | Atkinsom Index |
| Total effect | -0.012*** | -0.011** * | -0.008** | -0.005** | -0.006 | -0.004 |
| Structural effects | -0.015*** | -0.017*** | -0.024 | -0.027 | -0.021 | -0.019 |
| Characteristic effects | -0.007*** | -0.006*** | -0.002** | -0.004** | -0.003* | -0.002* |
| Measurement errors | -0.000 | -0.000 | -0.000 | -0.000 | -0.000 | -0.000 |
| Re-weighting errors | -0.004 | -0.005 | -0.003 | -0.004 | -0.003 | -0.005 |
| Re-weighting errors | | | | | | |
| Digital financial inclusion Index | -0.009*** | -0.007*** | -0.005** | -0.006** | -0.011 | -0.013 |
| Technological progress | -0.013*** | -0.014*** | -0.009 | -0.008 | -0.001 | -0.001 |
| Financial resource allocation capacity | -0.007*** | -0.006*** | -0.002 | -0.002 | -0.003 | -0.003 |
| Characteristic effects | | | | | | |
| Digital financial inclusion Index | -0.009*** | -0.011*** | -0.007** | -0.006** | -0.002* | -0.001* |
| Technological progress | -0.017*** | -0.015*** | -0.002** | -0.001** | -0.003 | -0.003 |
| Financial resource allocation capacity | -0.009*** | -0.007*** | -0.011** | -0.009** | -0.003* | -0.001* |

**2**

| | Central and West | | Central and Northeastern | | West and Northeastern | |
|---|---|---|---|---|---|---|
| | Dagum Gini coefficient | Atkinsom Index | Dagum Gini coefficient | Atkinsom Index | Dagum Gini coefficient | Atkinsom Index |
| Total effect | -0.001** | -0.002** | -0.001 | -0.003 | -0.002 | -0.003 |
| Structural effects | -0.002** | -0.003** | -0.004 | -0.007 | -0.004 | -0.002 |
| Characteristic effects | -0.011 | -0.013 | -0.002* | -0.003* | -0.001* | -0.002* |
| Measurement errors | -0.000 | -0.000 | -0.000 | -0.000 | -0.000 | -0.000 |
| Re-weighting errors | -0.002 | -0.001 | -0.002 | -0.001 | -0.002 | -0.003 |
| Re-weighting errors | | | | | | |
| Digital financial inclusion Index | -0.004** | -0.005** | -0.003 | -0.004 | -0.003 | -0.005 |
| Technological progress | -0.001** | -0.001** | -0.004 | -0.001 | -0.001 | -0.003 |
| Financial resource allocation capacity | -0.003** | -0.002** | -0.007 | -0.013 | -0.011 | -0.009 |
| Characteristic effects | | | | | | |
| Digital financial inclusion Index | -0.004 | -0.003 | -0.001* | -0.002* | -0.002* | -0.003* |
| Technological progress | -0.001* | -0.002* | -0.022 | -0.005 | -0.003 | -0.007 |
| Financial resource allocation capacity | -0.002 | -0.003 | -0.001* | -0.002* | -0.003* | -0.001* |

Note

***, **, and * represent significant at the 1%, 5%, and 10% levels respectively.

region than in others. However, the government's financial resource allocation ability can provide some funds for digital infrastructure construction to facilitate its agricultural transformation, empowering digital technology and digital financial knowledge in agriculture in the central region and narrowing the gap in agricultural carbon emission intensity between the central and east regions.

The total effect and structural effect of the eastern-western region are not significant. The decomposition coefficient of the core explanatory variable of the characteristic effect is significantly negative at the 10% level. Among the intermediate variables, only the financial resource

allocation ability is very harmful, and technological progress cannot reduce the Dagum Gini coefficient of agricultural carbon emission intensity between the eastern and western regions. There has long been a significant agricultural industrial structure, intensification, and commercialization gap between China's eastern and western regions. Digital financial inclusion can provide more convenient and inclusive financing channels for the development of large-scale agriculture and ecological environment improvement in the Western region, to some extent reducing the risk of agricultural green transformation and decreasing the agricultural carbon emission intensity in the Western region, achieving the convergence of carbon emission intensity gap between the eastern and western regions.

The total effect and structural effect of the central and western regions are significantly negative ($p<0.5$). In contrast, the decomposition coefficients of the core explanatory variable and intermediate variables (except technological progress) of the characteristic effect are insignificant. This indicates that the total effect has reduced the agricultural carbon emission intensity gap between the central and western regions. In contrast, the structural impact is the main factor in achieving convergence of the gap. Reducing agricultural carbon emission intensity between the west and central regions depends more on local financial resource allocation ability ($p<0.1$). This may be closely related to the relatively lagging construction of digital financial platforms and insufficient development momentum of digitalization and digital industrialization in the central and western regions. In the short term, eliminating the digital gap between the two regions requires the support of government financial policies.

The results for the central-northeast and western-northeast regions are consistent. The decomposition coefficients of the core explanatory variable and intermediate variables of the characteristic effect are significantly negative at the 10% level. However, the decomposition coefficients of the total and structural effects are insignificant. Among the intermediate variables, the ability to allocate financial resources plays a significant role, while technological progress has no significant impact.

## 6. Conclusions and countermeasures

### 6.1 Conclusions

This article analyzes the impact and mechanism of inclusive finance on the agricultural carbon emission intensity value and regional carbon emission intensity Dagum Gini coefficient of 30 sample provinces from 2010 to 2020. The study can provide relevant policy support for the current stage of digital inclusion to reduce agricultural carbon emissions, improve digital financial inclusion support for the transformation and upgrading of the regional agricultural industry, and provide feasible experiences for other countries. The following key conclusions are drawn from this study:

1. In general, digital financial inclusion can significantly reduce the carbon intensity of agriculture and promote convergence of the regional carbon emission intensity gap.

2. Based on the unconditional quantile regression coefficients, it is observed that the negative coefficients of the digital financial inclusion index and its three-dimensional indices decrease as the quantile increases. However, the significance of this effect varies significantly across different quantiles.

3. Technological progress and the government's financial resource allocation ability significantly mediate, and the urban-rural income gap further reduces the carbon intensity gap between regions.

## 6.2 Countermeasures

To address the regional gap in agriculture carbon emission, based on the development of digital financial inclusion and experimental research results, the following recommendations are suggested:

1. The Central Government should strengthen the construction and promotion of digital financial service platforms in rural and other economically disadvantaged areas, based on top-level policy design, to address the gap in infrastructure development and achieve equal access to inclusive digital finance services between urban and rural areas and among different regions. This will promote the development of agricultural industrialization, scale, and mechanization and improve the efficiency of resource and energy utilization. In addition, comprehensive efforts should be made to reduce the use of pesticides and fertilizers and to promote coordinated reduction of pollution and carbon emissions to achieve an overall decrease in agriculture carbon emission intensity and convergence in the regional carbon emission intensity gap.

2. Local governments should use digital financial inclusion as a new channel to support the local agriculture sector's green and low-carbon transformation. By utilizing information platforms, such as big data and cloud computing, local governments can develop agricultural models suitable for local conditions. Digital financial inclusion can help to break the unequal distribution of resources caused by traditional financial services, thus reducing the opportunity cost of low-carbon transformation in high-carbon emission regions. With high-quality financial services, digital financial inclusion can promote deep integration between regional agriculture carbon emission reduction and regional economic structure optimization.

3. Governments at all levels should strive to narrow the "digital divide" between regions by sharing resources, information, technology, and various types of capital through cooperation. This can be achieved by aggregating and spilling over financial elements and inclusive digital finance products to enrich the funding sources of different groups with financial needs and to seamlessly connect financial services with agriculture carbon emission reduction in other regions. High-carbon emission regions should be encouraged to seize opportunities for agricultural green transformation through the development of inclusive digital finance.

## 6.3 Innovations and limitations

Research on the impact of finance on regional differences in carbon emissions has a long history. Whether the digital transformation of agriculture promoted by digital financial inclusion can effectively achieve the dual-carbon goal of agriculture in the region, and whether it can reduce the intensity of agricultural carbon emissions in neighboring regions by means of the convenience, inter-temporality and low-cost characteristics of digital financial inclusion has not yet been validated, and even less has been demonstrated to show how digital financial inclusion can achieve convergence of the inter-regional disparity in agricultural carbon emissions. In view of this, this paper utilizes the panel data of 30 provinces in China (excluding Tibet, Hong Kong, Macao and Taiwan) from 2010 to 2020, and combines with Peking University's digital financial inclusion database, etc., in order to verify the impact of digital financial inclusion on the gap of inter-regional agricultural carbon emission intensity and the mechanism of its effect. The innovations in this study are as follows.

1. It examines the impact of digital financial inclusion on the intensity of agricultural carbon emission reduction. It investigates whether it can facilitate the reduction of agricultural carbon emissions and the convergence of the regional gap in agricultural carbon emission intensity.

2. It elucidates the impact mechanism of digital financial inclusion from financial resource allocation efficiency and technological innovation, offering a novel perspective for analyzing the path of convergence of the regional gap in carbon emission intensity through digital financial inclusion.

3. It incorporates the urban-rural income disparity. It comprehensively analyzes how digital financial inclusion influences the gap in agricultural carbon emission reduction intensity from the perspective of changes in the urban-rural income disparity. However, we also acknowledge some limitations.

However, we also admit limitation: Due to the data availability, this paper only analyzes the values of agricultural and regional carbon emission intensity Dagum Gini coefficient from 2010 to 2020, the effects of digital financial inclusion on them, and the underlying mechanisms. Therefore, the timeliness of the data may be inadequate, and we hope that future research will be carried out with more updated data sources. So, the time series of data can be added in future research to enhance the effectiveness of the results.

## Author Contributions

**Supervision:** Xiaxuan Li.

**Visualization:** Nuolan Tian.

**Writing – original draft:** Lingzhi Tan.

**Writing – review & editing:** Huan Chen.

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
