## [Decision Letter · Decision Letter 0]

2 Aug 2023

PONE-D-23-19988Can digital inclusion finance converge the regional agricultural carbon emissions gap?PLOS ONE

Dear Dr. Tan,

Thank you for submitting your manuscript to PLOS ONE. After careful consideration, we feel that it has merit but does not fully meet PLOS ONE’s publication criteria as it currently stands. Therefore, we invite you to submit a revised version of the manuscript that addresses the points raised during the review process.

We look forward to receiving your revised manuscript.

Kind regards,

Xiaobao Yu

Academic Editor

PLOS ONE

Journal Requirements:

"This work was supported by the Chongqing Philosophy and Social Science Project through a grant awarded to TLZ (2021NDYB087)."

"This work was supported by the Chongqing Philosophy and Social Science Project （2021NDYB087) "

This work was supported by the Chongqing Philosophy and Social Science Project through a grant awarded to TLZ (2021NDYB087)."

Reviewers' comments:

Reviewer's Responses to Questions

**Comments to the Author**

1. Is the manuscript technically sound, and do the data support the conclusions?

Reviewer #1: Yes

Reviewer #2: Yes

2. Has the statistical analysis been performed appropriately and rigorously? 

Reviewer #1: Yes

Reviewer #2: Yes

3. Have the authors made all data underlying the findings in their manuscript fully available?

Reviewer #1: Yes

Reviewer #2: Yes

4. Is the manuscript presented in an intelligible fashion and written in standard English?

Reviewer #1: Yes

Reviewer #2: No

5. Review Comments to the Author

Reviewer #1: Agricultural carbon intensity would be analyses with the quantile method in this article. I would suggest publishing the article as it is because it is provided sufficient evidence to investigate the occasion.

Reviewer #2: 1.Introduction: please start your elaboration from a much more macro perspective or background; for instance, you can illustrate the importance in a global background rather than only China.

2.You should update your references. I can see so many references from journals in China. More WoS journals published in the latest 5 years are needed. Some suggestions for you to cite; for instance, when talking about finance and economy: “Higher education and digital Economy: Analysis of their coupling coordination with the Yangtze River economic Belt in China as the example”; when talking about rural development: “Rural revitalization of China: A new framework, measurement and forecast (https://www.sciencedirect.com/science/article/pii/S0038012123002082)”; when talking about the importance to sustainability: “Research progress analysis of sustainable smart grid based on CiteSpace”; when talking about carbon emissions and environment: “Temporal-spatial measurement and prediction between air environment and inbound tourism: Case of China”.

3.Redraw figure 1: some lines seem wrong or improper.

4.Should model setting be placed before the descriptive statistical analysis? Please check whether current order is proper.

5.Refine titles. Some titles are too long; some are not proper; for instance, in 5.4, “etc.” is not commonly used in titles.

6.Section 6: put implications first, and then put conclusions. Also demonstrate your novelties and limitations.

7.Language must be polished. Current demonstration is Chinglish.

6. PLOS authors have the option to publish the peer review history of their article (what does this mean?). If published, this will include your full peer review and any attached files.

Reviewer #1: No

Reviewer #2: No

---

## [Author Response · Author response to Decision Letter 0]

18 Nov 2023

Dear Editors and Reviewers:

Thank you for your letter and for the reviewers’ comments concerning our manuscript entitled “Can digital financial inclusion converge the regional agricultural carbon emissions gap?” Those comments are all valuable for revising and improving our paper, as well as in favor of guiding our research.

We have accepted the reviewers’ important recommendations and strive for the best efforts to improve the manuscript. Specifically, we have made some revisions to the manuscript, but these changes will not affect the main content and framework of this paper. Revised portion are marked with red fonts in the paper. The main corrections in the paper and the responds to the reviewers’ comments are as flowing.

We appreciate for Editors/Reviewers’ warm work earnestly, and hope that the correction will meet with approval.

Once again, thank you very much for your comments and suggestions.

Sincerely yours,

Lingzhi Tan

 

Responds to the Editor's comments:

https://journals.plos.org/plosone/s/file?id=ba62/PLOSOne_formatting_sample_title_authors_affiliations.pdf: Thanks for the editor's comments. We apologize for the formatting errors in the initial draft. At the time of submission, we only had access to one file outlining the formatting requirements, which led to the errors. We have since made revisions to the draft in strict accordance with the journal's formatting standards.

2. Please state what role the funders took in the study. Please include this amended Role of Funder statement in your cover letter; we will change the online submission form on your behalf. 

Response: Thanks for the editor's comments. We have modified and added this section to the cover letter, as shown in the attached document.

3. Please remove any funding-related text from the manuscript and let us know how you would like to update your Funding Statement. Please include your amended statements within your cover letter.

Response: Thanks for the editor's comments. We have removed any funding-related text from the manuscript. And we have updated our funding statement in the cover letter, as shown in the attached document.

4. Please ensure that you have an ORCID iD and that it is validated in Editorial Manager.

Response: Thanks for the editor's comments. Apologies for our oversight, the ORCID ID has now been updated.

5. Please review your reference list to ensure that it is complete and correct.

Thanks for the editor's comments. After examining the literature, we found that none of the references in the manuscript had been retracted. However, we suspect that the issue raised by the editor may be due to the fact that several of the cited documents are in Chinese. Therefore, we have updated some of the references in the paper to incorporate their content.

The corresponding contents of the revised manuscript are as follows:

Pang J, Li H, Lu C, Lu C, Chen X. Regional Differences and Dynamic Evolution of Carbon Emission Intensity of Agriculture Production in China. International Journal of Environmental Research and Public Health. 2020; 17(20):7541. doi:10.3390/ijerph17207541

Wang L, Tang J, Tang M, Su M, Guo L. Scale of Operation, Financial Support, and Agricultural Green Total Factor Productivity: Evidence from China. Int J Environ Res Public Health. 2022;19(15):9043. Published 2022 Jul 25. doi:10.3390/ijerph19159043

Zhang F, Deng X, Phillips F, Fang CL, Wang C. Impacts of industrial structure and technical progress on carbon emission intensity: Evidence from 281 cities in China. Technological Forecasting & Social Change. 2020; 154:435-447.doi: 10.1016/j.techfore.2020.119949

Wei P, Xiong LY. Managing financing costs and fostering green transition: The role of green financial policy in China. Economic Analysis and Policy.2022;76:820-836. doi: 10.1016/J.EAP.2022.09.014

Lyu XY. The Impact of the Digital Economy on Carbon Emissions: Evidence from China. Journal of Empirical Studies. 2022; 9:15-23. doi:10.18488/66. v9i1.3066.

Hong M, Tian M, Wang J. Digital financial inclusion, Agricultural Industrial Structure Optimization and Agricultural Green Total Factor Productivity. Sustainability. 2022; 14(18):11450. doi: 10.3390/su141811450

Sun YN, You XT. Do digital financial inclusion, innovation, and entrepreneurship activities stimulate vitality of the urban economy? Empirical evidence from the Yangtze River Delta, China. Technology in Society. 2023; 72. doi: 10.1016/j.techsoc.2023.102200

Liu Y, Luan L, Wu WL, Zhang ZQ, Hsu Y. Can digital financial inclusion promote China's economic growth? International Review of Financial Analysis.2021;78:101889. doi: 10.1016/J.IRFA.2021.101889

Wang WJ, He TY, Li ZH. Digital financial inclusion, economic growth and innovative development. Kybernetes. 2022. doi:10.1108/K-09-2021-0866. 

6.Please ensure that you refer to Figure 1 in your text as, if accepted, production will need this reference to link the reader to the figure.

Response: Thanks for the editor's comments. We have re-referenced Figure 1 in the text.

Responds to the reviewer’s comments:

Reviewers’ comments:

Reviewer #1: Agricultural carbon intensity would be analyses with the quantile method in this article. I would suggest publishing the article as it is because it is provided sufficient evidence to investigate the occasion. 

Response: Thanks for the reviewer’s comments. It is our honor to receive your approval. 

Reviewer #2:

1. Introduction: please start your elaboration from a much more macro perspective or background; for instance, you can illustrate the importance in a global background rather than only China.

Response: Thanks for the reviewer’s comments. Your feedback was very helpful, as it specifically highlighted issues with the introduction section of our paper. As a result, we have made revisions to the introduction in order to emphasize the global significance of the topic, rather than solely focusing on China.

The corresponding contents of the revised manuscript are as follows:

The former establishment of a green, low-carbon, and sustainable global governance system has become a common development goal for all countries. Agricultural production is strongly affected by and a major contributor to climate change. Agriculture and land-use change account for a quarter of total global emissions of greenhouse gases (GHG). In the 20-year period, China was responsible for the most emissions from agricultural production. Consequently, developing LCA with "high efficiency, low energy consumption, low emissions, and high carbon sinks" is crucial. 

2. You should update your references. 

Response: Thanks for the reviewer’s comments. We have read the references provided by you and found them to be very useful in enhancing our paper. Therefore, we have cited all of them in the manuscript. For details, please refer to the revised manuscript.

3. Redraw figure 1: some lines seem wrong or improper. 

Response: Thanks for the reviewer’s comments. We identified some issues with Figure 1 and have reworked it accordingly. For details, please refer to the revised manuscript.

4. Should model setting be placed before the descriptive statistical analysis? Please check whether current order is proper.

Response: Thanks for the reviewer’s comments. We have placed the model setting before the descriptive statistics analysis. For details, please refer to the revised manuscript.

5. Refine titles. Some titles are too long; some are not proper; for instance, in 5.4, “etc.” is not commonly used in titles. 

Response: Thanks for the reviewer’s comments. We reread the article and made simplifying changes to the headline.

The corresponding contents of the revised manuscript are as follows:

2.3 Mechanism of action

2.4 Moderating effect

4.3 Mechanism analysis

4.4 Moderating effect

5.4 Decomposition of interregional disparities

6. Conclusions and countermeasures

6.1 Conclusions

6.2 Countermeasures

6. Section 6: put implications first, and then put conclusions. Also demonstrate your novelties and limitations.

Response: Thanks for the reviewer's comments. We have revised the Section 6 by first presenting the implications, followed by the conclusions and countermeasures. Additionally, we have included a discussion of the novelties and limitations of our paper at the end.

The corresponding contents of the revised manuscript are as follows:

This article analyzes the impact and mechanism of inclusive finance on the agricultural carbon emission intensity value and regional carbon emission intensity D-Gini coefficient of 30 sample provinces from 2010 to 2020. The study can provide relevant policy support for the current stage of digital inclusion financial to reduce agricultural carbon emissions and improve digitalinclusive finance support for the transformation and upgrading of the regional agricultural industry, as well as provide feasible experiences for other countries. We get the following key conclusions in this study:

6.3 Innovations and limitations

The innovations in this study are as follows.

1. It analyzes the impact of digital financial inclusion on agricultural carbon emission reduction intensity to determine whether it can help reduce agricultural carbon emissions and narrow the gap in agricultural carbon emission intensity between regions. 

2. It explains the impact mechanism of digital financial inclusion from the perspectives of financial resource allocation capacity and technological progress, providing a new perspective for analyzing the path of convergence of the regional carbon emission intensity gap through digital financial inclusion. 

3. It introduces the urban-rural income gap and provides an in-depth analysis of how digital financial inclusion affects the gap in agricultural carbon emission reduction intensity from the perspective of changes in the urban-rural income gap.

However, we also admit some limitations.

1. Limited by the research data, this paper only analyses the value of agricultural carbon emission intensity and regional carbon emission intensity D-Gini coefficient from 2010 to 2020, the results of digital financial inclusion on the two and the mechanism of action. Therefore, the timeliness of the data may be insufficient, and it is hoped that further research will be conducted in the future using updated data sources.

2. We cited more Chinese literature, mainly considering that this academic language has extensively studied issues related to this article, but documents in other languages may also be beneficial.

7. Language must be polished. Current demonstration is Chinglish.

Response: Thanks for the reviewer's comments. We have polished the language of the article and hope that the issues you raised will be improved. For details, please refer to the revised manuscript.

---

## [Decision Letter · Decision Letter 1]

8 Jan 2024

PONE-D-23-19988R1Can digital financial inclusion converge the regional agricultural carbon emissions gap?PLOS ONE

Dear Dr. Tan,

Thank you for submitting your manuscript to PLOS ONE. After careful consideration, we feel that it has merit but does not fully meet PLOS ONE’s publication criteria as it currently stands. Therefore, we invite you to submit a revised version of the manuscript that addresses the points raised during the review process.

We look forward to receiving your revised manuscript.

Kind regards,

Xiaobao Yu

Academic Editor

PLOS ONE

Reviewers' comments:

Reviewer's Responses to Questions

**Comments to the Author**

1. If the authors have adequately addressed your comments raised in a previous round of review and you feel that this manuscript is now acceptable for publication, you may indicate that here to bypass the “Comments to the Author” section, enter your conflict of interest statement in the “Confidential to Editor” section, and submit your "Accept" recommendation.

Reviewer #3: All comments have been addressed

Reviewer #4: (No Response)

2. Is the manuscript technically sound, and do the data support the conclusions?

Reviewer #3: No

Reviewer #4: Yes

3. Has the statistical analysis been performed appropriately and rigorously? 

Reviewer #3: No

Reviewer #4: Yes

4. Have the authors made all data underlying the findings in their manuscript fully available?

Reviewer #3: No

Reviewer #4: Yes

5. Is the manuscript presented in an intelligible fashion and written in standard English?

Reviewer #3: No

Reviewer #4: No

6. Review Comments to the Author

Reviewer #3: I have taken note of the following points that need to be addressed:

- The reference style needs to be corrected to focus only on the family name. Additionally, all citations in the text should be updated in the reference.

- There are many typos throughout the paper, and the English needs to be polished. The authors sometimes use past tense and sometimes present tense, so this needs to be revised.

- In the abstract, "D-GINI" should be written as the full word at the beginning where it first appears.

- The contribution and research gap in this study needs to be mentioned in the last paragraph to highlight the novelty of the paper.

- The term "digitalinclusive finance" is mentioned in the abstract but only once in the literature review. It is unclear if "digital financial inclusion" and "digitalinclusive finance" are the same thing or not. The authors should clarify this and also consider focusing on digitalinclusive finance and CO2 emissions.

- There are errors in lines 324, 496, and 528.

- The Dagum Gini coefficient is not explained in section 3.1 and needs to be defined.

- The methodology section is problematic as it mixes up everything and is not explained clearly.

- Eqs (1-3) in the methodology section are not clear, and the authors should explain whether they represent the same model and why they have different structures.

- Eq (6) is unclear and the explanation below is confusing.

- Eqs (8-10) are also not clear, and many terms are not explained. The authors should define δ.

- Section 3.2.4 is not clear, and the RIF-Blinder-Oaxaca decomposition method needs to be explained as well as how to decompose it.

- It is unclear whether Table 3 was produced from RIF models and which equations were used.

- Table 5 is derived from quantile regression, but the explanation in the methodology section does not make sense. All the coefficients are constant and do not vary according to quantile level.

- Overall, the paper looks messy and is not yet ready to be published.

Reviewer #4: 1. The title solely addresses the carbon emission gap, without acknowledging carbon emission intensity. Nevertheless, it is crucial to note that the text extensively covers the aspect of carbon emission intensity as well. Hence, I suggest authors focus on the carbon emission intensity may be more suitable. You can refer to the following papers related to carbon emission intensity: Does digital inclusive finance affect urban carbon emission intensity: Evidence from 285 cities in China. Cities

2. The revision of regional differences into provincial differences is recommended.

3. The conflicting arguments regarding carbon intensity and its regional disparities may potentially perplex the readers.

4. The methodology section suggests emphasizing the merits of the RIF method in coping with carbon intensity gap measurements.

5. The presence of numerous errors is evident. Such as “Error! Reference source not found.”

6. The excessive inclusion of Chinese references in the paper does not align with the criteria set by English journals. I sugges authors cite more papers from international journals. Some papers maybe useful for your revision: Seeing green: how does digital infrastructure affect carbon emission intensity? Energy Economics. Reducing carbon emissions: Can high-speed railway contribute? Journal of Cleaner Production.

7. The language expression and textual logic still have ample room for improvement. Take the first paragraph of the introduction for example, what does “The former establishment” refer to? What does “the 20-year period” mean? What does “Agriculture and land-use change account for a quarter of total global emissions of greenhouse gases” mean? What does “LCA” refer to?

7. PLOS authors have the option to publish the peer review history of their article (what does this mean?). If published, this will include your full peer review and any attached files.

Reviewer #3: No

Reviewer #4: No

---

## [Author Response · Author response to Decision Letter 1]

23 Jan 2024

Dear Editors and Reviewers:

Thank you for your letter and the reviewers' comments concerning our manuscript entitled "Can digital financial inclusion converge the regional agricultural carbon emissions intensity gap?" Those comments are all valuable for revising and improving our paper and help guide our research.

We have accepted the reviewers' important recommendations and strive to make the best effort to improve the manuscript. Specifically, we have made some revisions to the manuscript, but these changes will not affect this paper's leading content and framework. Revised portions are marked with red fonts in the paper. The main corrections in the paper and the responds to the reviewers' comments are as flowing.

We appreciate for Editors/Reviewers' warm work earnestly, and hope that the correction will meet with approval.

Once again, thank you very much for your comments and suggestions.

Sincerely yours,

Lingzhi Tan

 

Responds to the reviewer's comments:

Reviewers' comments:

Reviewer #3: 

1. The reference style needs to be corrected to focus only on the family name. Additionally, all citations in the text should be updated in the reference. 

Response: Thanks for the reviewer's comments. We have also noticed this problem. Therefore, we have updated the references in a timely manner according to the journal formatting requirements. At the same time, all citations in the text were updated in the references accordingly.

2. There are many typos throughout the paper, and the English needs to be polished. The authors sometimes use past tense and sometimes present tense, so this needs to be revised.

Response: Thank you for your suggestions. They help us a lot to improve the quality of our articles. We have re-polished the language of the article. 

3. In the abstract, "D-GINI" should be written as the full word at the beginning where it first appears.

Response: Thanks for the reviewer's comments. This was indeed an oversight, and we have now changed the first occurrence of the word "D-GINI" to the whole word.

3. The contribution and research gap in this study needs to be mentioned in the last paragraph to highlight the novelty of the paper.

Response: Thanks for the reviewer's comments. Section 6.3 of the article presents the contribution and research gap in this study as follows:

The innovations in this study are as follows.

1. It examines the impact of digital financial inclusion on the intensity of agricultural carbon emission reduction. It investigates whether it can facilitate the reduction of agricultural carbon emissions and the convergence of the regional gap in agricultural carbon emission intensity.

2. It elucidates the impact mechanism of digital financial inclusion from financial resource allocation efficiency and technological innovation, offering a novel perspective for analyzing the path of convergence of the regional gap in carbon emission intensity through digital financial inclusion.

3. It incorporates the urban-rural income disparity. It comprehensively analyzes how digital financial inclusion influences the gap in agricultural carbon emission reduction intensity from the perspective of changes in the urban-rural income disparity. However, we also acknowledge some limitations.

However, we also admit limitation: Due to the data availability, this paper only analyzes the values of agricultural and regional carbon emission intensity Dagum Gini coefficient from 2010 to 2020, the effects of digital financial inclusion on them, and the underlying mechanisms. Therefore, the timeliness of the data may be inadequate, and we hope that future research will be carried out with more updated data sources. So, the time series of data can be added in future research to enhance the effectiveness of the results.

5. The term "digital inclusive finance" is mentioned in the abstract but only once in the literature review. It is unclear if "digital financial inclusion" and "digital inclusive finance" are the same thing or not. The authors should clarify this and also consider focusing on digital inclusive finance and CO2 emissions.

Response: Thanks for the reviewer's comments. The issue you mentioned was an oversight in our writing and has now been changed to "digital financial inclusion."

6. There are errors in lines 324, 496, and 528. 

Response: Thanks for the reviewer's comments. Many thanks to the reviewers for their meticulous review, which will immensely help our article. The errors have now been corrected and can be viewed in the main text.

7. The Degum Gini coefficient is not explained in section 3.1 and needs to be defined.

Response: Thanks for the reviewer's comments. The Degum Gini coefficient has been explained in Section 3.1 by defining it as follows:

In addition, the Dagum method was used to measure the differences in agricultural carbon emission intensity between regions in China. At this stage, the indicators for studying provincial differences mainly include the coefficient of variation, Mean Log Deviation Index (MLD), Tyrell's index, and Gini coefficient. However, when analyzing the above methods, most of them cannot decompose the provincial differences or do not consider the distribution of sample subgroups when deteriorating, so they all have certain defects in the application. To overcome the above difficulties, some researchers have used Dagum Gini coefficient to analyze provincial dynamic differences and evolutionary trends. In this paper, the Dagum Gini coefficient is utilized to measure the inter-provincial differences in the intensity of agricultural carbon emissions in China and decompose them to obtain the composition of the relative differences between inter-provincial regions.

G=∑_(j=1)^k▒∑_(h=1)^k▒∑_(i=1)^(C_j)▒∑_(r=1)^(C_h)▒| y_ji-y_hr |/2C^2 y ®（6）

In the equation, yji (yhr) denotes the agricultural carbon intensity of any provincial unit within the j(h) region, C is the number of sample provinces, is the national average of agricultural carbon intensity, k is the number of regional divisions, and Cj(Ch) is the number of provincial units within the j(h) region.

G_jh=∑_(i=1)^(G_j)▒∑_(r=1)^(G_h)▒| y_ji-y_hr |/C_j C_h (Y ®_j+Y ®_h)（7）

Equation (7) is the Dagum Gini coefficient of agricultural carbon intensity between provinces j and h. Y ®_j and Y ®_h are the average agricultural carbon emission values of provinces j and h.

7. The methodology section is problematic as it mixes up everything and is not explained clearly.

Response: Thanks for the reviewer's comments. The method formulas utilized in the text are shown in the hope of answering your questions. The method in the text is specifically:

1.Baseline model

According to the research objectives and previous literature, we constructed the following RIF regression model:

RIF{〖lny〗_it,v^(D-G) (F_Y)}=β_0+β_1 X_(it-1)+β_2 Z_(it-1)+∅+η+ε_it（1）

RIF(lny_it;q_r,F)=q_r+(τ-I(lny_(it-1)≤q_r))/(f_lny (q_r))（2）

RIF(lny_it│∅;γ;q_r )=BX ~(Z|) ~∅+ε_it（3）

To verify the establishment of H1, the base regression model adopts the regression method of regrouping influence function (RIF), as shown in Eq (1); to consider the changes in the intensity of interprovincial agricultural carbon emission reductions at different quartiles, an unconditional quantile regression model is constructed based on the decentralized mapping function shown in Eq (2), as shown in Eq (3). In Eqs. (1)-(3), 〖lny〗_it denotes the value of agricultural carbon intensity in region i in year t, and lny_(it-1) is the value of lagged one-period carbon intensity. The term v^(D-G) (F_Y) indicates the Dagum Gini coefficient of lny, which reflects the degree of imbalance in carbon intensity in province i in year t. X represents the digital financial inclusion index, further divided into the breadth of coverage, depth of use, and degree of digital services. X_(it-1) is the lagged one-period index value for the sample provinces. β_0 is the intercept term, β_1and β_2 are the regression coefficient and Z_(it-1) is a set of macro control variables lagged by one period i denotes the sample province, and t represents the year. ∅ signifies province and industry fixed effects, η refers to time-fixed effects, and ε_it is a random disturbance term. τ represents quantile points, q_r is the unconditional quantile function of lny, f_lny (q_r) is the probability density function, and B is the vector matrix of regression coefficients. X ~and Z ~ are the vector matrices of the core explanatory and control variables.

2. Calculation of carbon intensity in agriculture

Referring to the 2006 National Greenhouse Gas Inventory Guidelines Carbon Emission Factor Methodology recommended by the IPCC: 

TC=∑▒〖C_i=∑▒〖S_i×ρ_i 〗〗（4）

Where TC is the total agricultural carbon emissions, C_i denotes the total agricultural carbon emissions of the ith carbon emission source, S_i refers to the total carbon emission input of the ith carbon emission source, and ρ_i represents the emission factor of each carbon emission source. The reference coefficients and literature sources used in the equation are presented in Table 2.

 Table 2 Carbon emission factors for primary carbon sources 

Carbon source factor Carbon emission factor Literature sources

Fertilizer 0.896 kgC .kg-1 Oak Ridge National Laboratory

Pesticides 4.934 kgC .kg-1 Oak Ridge National Laboratory

Agricultural film 5.180 kgC .kg-1 The Institute of Resource, Ecosystem, and Environment of Agriculture (IREEA) of Nanjing Agricultural University 

Diesel 0.593 kgC.kg-1 Intergovernmental Panel on Climate Change 

Ploughing 312.603 kgC.kg-1 Wu FL et al.（2008）[50] 

Irrigation 20.476 kgC.kg-1 Duan HP et al.（2012）[51]

Calculate the value of agricultural carbon intensity according to equation (4):

E_agri=〖TC〗_it |〖GDP〗_it（5）

In the equation, E_agri represents the value of agricultural carbon emission intensity (KG/10000). GDP refers to the total agricultural output value of the sample provinces, and both variables are included in the equation using logarithmic values.

3. Calculation of inter-provincial differences in agricultural carbon intensity

In addition, the Dagum method was used to measure the differences in agricultural carbon emission intensity between regions in China. At this stage, the indicators for studying provincial differences mainly include the coefficient of variation, Mean Log Deviation Index (MLD), Tyrell's index Gini coefficient, etc. However, when analyzing the above methods, most of them cannot decompose the provincial differences or do not consider the distribution of sample subgroups when deteriorating, so they all have certain defects in the application. To overcome the above difficulties, some researchers have used Dagum Gini coefficient to analyze provincial dynamic differences and evolutionary trends. In this paper, the Dagum Gini coefficient is utilized to measure the inter-provincial differences in the intensity of agricultural carbon emissions in China and decompose them to obtain the composition of the relative differences between inter-provincial regions.

G=∑_(j=1)^k▒∑_(h=1)^k▒∑_(i=1)^(C_j)▒∑_(r=1)^(C_h)▒| y_ji-y_hr |/2C^2 y ®（6）

In the equation, yji (yhr) denotes the agricultural carbon intensity of any provincial unit within the j(h) region, C is the number of sample provinces, is the national average of agricultural carbon intensity, k is the number of regional divisions, and Cj(Ch) is the number of provincial units within the j(h) region.

G_jh=∑_(i=1)^(G_j)▒∑_(r=1)^(G_h)▒| y_ji-y_hr |/C_j C_h (Y ®_j+Y ®_h)（7）

Equation (7) is the Dagum Gini coefficient of agricultural carbon intensity between provinces j and h. Y ®_j and Y ®_h are the average agricultural carbon emission values of provinces j and h.

9. Eqs (1-3) in the methodology section are not clear, and the authors should explain whether they represent the same model and why they have different structures.

Response: Thanks for the reviewer's comments. To verify the establishment of H1, the base regression model adopts the regression method of regrouping influence function (RIF), as shown in Eq (1); to consider the changes in the intensity of interprovincial agricultural carbon emission reductions at different quartiles, an unconditional quantile regression model is constructed based on the decentralized mapping function shown in Eq (2), as shown in Eq (3).

10. Eq (6) is unclear and the explanation below is confusing.

Response: Thanks for the reviewer's comments. I will re-explain Eq (6) here and hopefully answer your questions. At this stage, the indicators for studying provincial differences mainly include coefficient of variation, Mean Log Deviation Index (MLD), Tyrell's index and Gini Coefficient, etc. However, when analyzing the above methods, most of them cannot decompose the provincial differences or do not consider the distribution of sample subgroups when decomposing, so they all have certain defects in the application. To overcome the above difficulties, some researchers have used Dagum Gini coefficient to analyze provincial dynamic differences and evolutionary trends. In this paper, the Dagum Gini coefficient is utilized to measure the inter-provincial differences in the intensity of agricultural carbon emissions in China and decompose them to obtain the composition of the relative differences between inter-provincial regions.

G=∑_(j=1)^k▒∑_(h=1)^k▒∑_(i=1)^(C_j)▒∑_(r=1)^(C_h)▒| y_ji-y_hr |/2C^2 y ®（6）

In the equation, yji (yhr) denotes the agricultural carbon intensity of any provincial unit within the j(h) region, C is the number of sample provinces, is the national average of agricultural carbon intensity, k is the number of regional divisions, and Cj(Ch) is the number of provincial units within the j(h) region.

11. Eqs (8-10) are also not clear, and many terms are not explained. The authors should define δ.

Response: Thanks for the reviewer's comments. I'm going to re-explain Eqs (8-10) here and hopefully answer your questions:

To test hypothesis H3, a mediating effects model was used to test the mediating effects of technological progress, the government's ability to allocate financial resources, and the mediating effects of the combination of the two, respectively.

RIF{M_n;v^DG (F_Y)}=γ_0+γ_nX+λ_2 Z+∅+μ （8）

RIF{〖ln〗_(y_it ),v^DG (F_Y)}=k_0+∑_(n=1)^2▒〖φ_n M_n+k_1 X_it+k_2 Z_(it-1)+∅+δ〗(9)

β_1=k_1+∑_(n=1)^2▒〖o_n γ_n 〗(10)

In the equation, μ and δ are random disturbance terms, M_n the mediating variable includes technological progress and financial resource allocation capacity. γ_0 and k_0 are intercept terms. γ_nand λ_2 are regression coefficients, k_1 denotes the direct effect of the core explanatory variables, and β_1 is the total effect of the impact of digital financial inclusion on the intensity of carbon reduction in agriculture and the gap in the intensity of carbon reduction in agriculture. φ_n and γ_n refer to individual mediating effects, and β_1- k_1 signifies the joint mediating effect. If γ_n in equation (8) and φ_n in equation (9) are both significant, it can be shown that digital financial inclusion can reduce agricultural carbon emission intensity and Dagum Gini coefficient through the mediating variable M_it. Conversely, if they are not significant, the opposite is true.

12. Section 3.2.4 is unclear, and the RIF-Blinder-Oaxaca decomposition method needs to be explained and how to decompose it.

Response: Thanks for the reviewer's comments. The RIF-Blinder-Oaxaca decomposition methodology and how it is done has been re-explained in the text as follows:

The regional agricultural carbon emission intensity gap decomposition method combines the implications of the regression of the regrouping influence function (RIF) method, which decomposes the total interprovincial agricultural carbon emission intensity differences into structural and characterization effects by constructing a counterfactual distribution function, so as to obtain the specific contribution of each explanatory variable to the total differences, and extends it to RIF regressions of the Gini coefficient and the Atkinson's index. Among them, the structural effect is determined by the difference in total agricultural carbon emissions between different inter-province caused by the different shares of agricultural GDP; the characterization effect is caused by the differences in the in

---

## [Decision Letter · Decision Letter 2]

29 Feb 2024

PONE-D-23-19988R2Can digital financial inclusion converge the regional agricultural carbon emissions intensity gap?PLOS ONE

Dear Dr. Tan,

Thank you for submitting your manuscript to PLOS ONE. After careful consideration, we feel that it has merit but does not fully meet PLOS ONE’s publication criteria as it currently stands. Therefore, we invite you to submit a revised version of the manuscript that addresses the points raised during the review process.

We look forward to receiving your revised manuscript.

Kind regards,

Xiaobao Yu

Academic Editor

PLOS ONE

Journal Requirements:

Reviewers' comments:

Reviewer's Responses to Questions

**Comments to the Author**

1. If the authors have adequately addressed your comments raised in a previous round of review and you feel that this manuscript is now acceptable for publication, you may indicate that here to bypass the “Comments to the Author” section, enter your conflict of interest statement in the “Confidential to Editor” section, and submit your "Accept" recommendation.

Reviewer #3: (No Response)

Reviewer #4: All comments have been addressed

Reviewer #5: (No Response)

2. Is the manuscript technically sound, and do the data support the conclusions?

Reviewer #3: No

Reviewer #4: Yes

Reviewer #5: (No Response)

3. Has the statistical analysis been performed appropriately and rigorously? 

Reviewer #3: No

Reviewer #4: Yes

Reviewer #5: (No Response)

4. Have the authors made all data underlying the findings in their manuscript fully available?

Reviewer #3: No

Reviewer #4: Yes

Reviewer #5: (No Response)

5. Is the manuscript presented in an intelligible fashion and written in standard English?

Reviewer #3: No

Reviewer #4: Yes

Reviewer #5: (No Response)

6. Review Comments to the Author

Reviewer #3: Some citations remain unedited, such as in Section 2.3 (e.g., Suhrab M et al. (2024), Chen YB et al. (2022)[43]).

I strongly suggest improving the discussion on the contribution and research gap in this study. It should be highlighted in the last paragraph to underscore the paper's novelty. Additionally, Section 6.3 only outlines the purpose and advantages of the study; there is no clear identification of the research gap or the unique value this paper adds to the existing literature.

The methodology is unclear and contains unusual notations. For instance, ^− () represents the Dagum Gini coefficient whixh doest not make any sense here, and some notations lack italic formatting. Equations used in this study are not convincing, and there is a missing explanation of the unconditional quantile regression model.

In Table 4, the differences between columns 1-2, 4-5, and 7-8 are unclear. How are they different.

I could provide more feedback, but there are numerous errors and unclear explanations. The paper appears messy and sloppy.

Reviewer #4: (No Response)

Reviewer #5: I am pleased to acknowledge that the revised version of the manuscript has been considerably enhanced in terms of its comprehensiveness and overall content.

7. PLOS authors have the option to publish the peer review history of their article (what does this mean?). If published, this will include your full peer review and any attached files.

Reviewer #3: No

Reviewer #4: No

Reviewer #5: **Yes: **Dr. Kishwar Ali

---

## [Author Response · Author response to Decision Letter 2]

14 Apr 2024

Dear Editors and Reviewers:

Thank you for your letter and the reviewers' comments concerning our manuscript entitled "Can digital financial inclusion converge the regional agricultural carbon emissions intensity gap?" Those comments are all valuable for revising and improving our paper and help guide our research.

We have accepted the reviewers' important recommendations and strive to make the best effort to improve the manuscript. Specifically, we have made some revisions to the manuscript, but these changes will not affect this paper's leading content and framework. Revised portions are marked with red fonts in the paper. The main corrections in the paper and the responds to the reviewers' comments are as flowing.

We appreciate for Editors/Reviewers' warm work earnestly, and hope that the correction will meet with approval.

Once again, thank you very much for your comments and suggestions.

Sincerely yours,

Lingzhi Tan

 

Responds to the reviewer's comments:

Reviewers' comments:

Reviewer #3: 

1.Some citations remain unedited, such as in Section 2.3 (e.g., Suhrab M et al. (2024), Chen YB et al. (2022) [43]).

Response: Thanks for the reviewer's comments. The problem you are referring to is the typography that is needed according to the article. However, we have partially taken your comments and corrected them.

2. The discussion on the contribution and research gap in this study. It should be highlighted in the last paragraph to underscore the paper's novelty. Additionally, Section 6.3 only outlines the purpose and advantages of the study; there is no clear identification of the research gap or the unique value this paper adds to the existing literature.

Response: Thanks for the reviewer's comments. Your comments were very helpful, so we have made the following changes:

Research on the impact of finance on regional differences in carbon emissions has a long history. Whether the digital transformation of agriculture promoted by digital financial inclusion can effectively achieve the dual-carbon goal of agriculture in the region, and whether it can reduce the intensity of agricultural carbon emissions in neighboring regions by means of the convenience, inter-temporality and low-cost characteristics of digital financial inclusion has not yet been validated, and even less has been demonstrated to show how digital financial inclusion can achieve convergence of the inter-regional disparity in agricultural carbon emissions. In view of this, this paper utilizes the panel data of 30 provinces in China (excluding Tibet, Hong Kong, Macao and Taiwan) from 2010 to 2020, and combines with Peking University's digital financial inclusion database, etc., in order to verify the impact of digital financial inclusion on the gap of inter-regional agricultural carbon emission intensity and the mechanism of its effect. The innovations in this study are as follows.

1. It examines the impact of digital financial inclusion on the intensity of agricultural carbon emission reduction. It investigates whether it can facilitate the reduction of agricultural carbon emissions and the convergence of the regional gap in agricultural carbon emission intensity.

2. It elucidates the impact mechanism of digital financial inclusion from financial resource allocation efficiency and technological innovation, offering a novel perspective for analyzing the path of convergence of the regional gap in carbon emission intensity through digital financial inclusion.

3. It incorporates the urban-rural income disparity. It comprehensively analyzes how digital financial inclusion influences the gap in agricultural carbon emission reduction intensity from the perspective of changes in the urban-rural income disparity. However, we also acknowledge some limitations.

However, we also admit limitation: Due to the data availability, this paper only analyzes the values of agricultural and regional carbon emission intensity Dagum Gini coefficient from 2010 to 2020, the effects of digital financial inclusion on them, and the underlying mechanisms. Therefore, the timeliness of the data may be inadequate, and we hope that future research will be carried out with more updated data sources. So, the time series of data can be added in future research to enhance the effectiveness of the results.

3.The methodology is unclear and contains unusual notations. For instance, ^− () represents the Dagum Gini coefficient whixh doest not make any sense here, and some notations lack italic formatting. Equations used in this study are not convincing, and there is a missing explanation of the unconditional quantile regression model.

Response: Thanks for the reviewer's comments. 

About the methodology. Firstly, this paper adopts the regression method of regrouping influence function (RIF) to elucidate the impact of digital financial inclusion on the intensity and variation of agricultural carbon emissions between provinces in China. The specific procedures are as follows: first, the conditional mean function, quantile points and Dagum Gini coefficient are used to conduct RIF regression analysis. Specifically, the conditional mean function is used for RIF regression analysis to test whether digital financial inclusion affects the change of interprovincial agricultural carbon emission intensity; the unconditional quantile regression model is constructed to consider the change of interprovincial agricultural carbon emission intensity at different quantile points; second, the mediation effect model is used to analyze the mechanism of the impact of digital financial inclusion on the regional disparity of agricultural carbon emission intensity; third, the RIF- Blinder-Oaxaca model to construct a counterfactual distribution function to decompose the influencing factors of the interprovincial agricultural carbon emission reduction intensity gap; and the moderating effect model is used to verify the influencing effects of the moderating variables. Second, the mediating effects model is used to test the mediating effects of technological progress and the government's ability to allocate financial resources, as well as the mediating effects of the combination of the two, respectively. Lastly, the moderating effect of the urban-rural income gap on the convergence of regional agricultural carbon emission reduction intensities was examined, while the RIF-Blinder-Oaxaca decomposition method was used to seek the main factors affecting the regional agricultural carbon emission reduction intensity gap.

Regarding the uncommon symbols and italics in the text, we have changed them in the text, thank you for your comments.

The unconditional quantile regression model is interpreted as follows:

The general measurement method cannot be easily extended to non-central locations, and only observes the average treatment effect of digital financial inclusion, ignoring the impact of digital financial inclusion on regional heterogeneity of inter-provincial agricultural carbon emission intensity, such as the impact of digital financial inclusion on high-agricultural carbon-emitting provinces (upper tail) versus low-agricultural carbon-emitting provinces (low-tail), which tend to be the two groups that are the most direct role of digital financial inclusion in carbon agricultural emission reduction effects groups.Firpo et al. (2018) further add an extension to the technique of conditional quantile regression with the proposed regression of the regrouping influence function (RIF), which assesses the effect of changes in the distribution of the explanatory variables on the quantiles of the unconditional (marginal) distributions of the outcome variables, and is able to achieve the important breakthrough of overstepping from focusing on decompositions of differences in means to decompositions of differences in the entire distribution, as well as being able to give the degree of influence of each independent variable, better avoiding the endogeneity problem arising from omitted variables. Based on this, the regression method of regrouping influence function (RIF) is used to elucidate the impact of digital inclusive finance on the changes of agricultural carbon emission intensity and differences between provinces in China.Firpo et al (2018) argued that the regression method of regrouping influence function can better avoid the problem of endogeneity generated by the omitted variables, and is currently mostly used in the income disparity, rich-poor disparity measurement and so on. The specific procedures are as follows: First, the conditional mean function, quantile points and Dagum Gini coefficient are used for RIF regression analysis. Specifically, the conditional mean function is used for RIF regression analysis to test whether digital financial inclusion affects the change of interprovincial agricultural carbon emission intensity; the unconditional quantile regression model is constructed to consider the change of interprovincial agricultural carbon emission intensity at different quantile points; second, the mediation effect model is used to analyze the mechanism of the impact of digital financial inclusion on the regional disparity of the agricultural carbon emission intensity; and third, the RIF- Blinder-Oaxaca model to construct a counterfactual distribution function to decompose the influencing factors of the interprovincial agricultural carbon emission reduction intensity gap; and the moderating effect model is used to verify the influencing effects of the moderating variables.

4.In Table 4, the differences between columns 1-2, 4-5, and 7-8 are unclear. How are they different.

Response: Thanks for the reviewer's comments. As explained earlier, To avoid your question, the explanation has now been reintroduced in the text as follows: The difference between them is that columns (1) (4) (7) are the conditional mean model without control variables, while columns (2) (5) and (8) are the regression results after adding the control variables.

---

## [Editor Report · Decision Letter 3]

17 Apr 2024

Can digital financial inclusion converge the regional agricultural carbon emissions intensity gap?

PONE-D-23-19988R3

Dear Dr. Tan,

We’re pleased to inform you that your manuscript has been judged scientifically suitable for publication and will be formally accepted for publication once it meets all outstanding technical requirements.

Kind regards,

Xiaobao Yu

Academic Editor

PLOS ONE

---

## [Editor Report · Acceptance letter]

4 Jul 2024

PONE-D-23-19988R3 

PLOS ONE

Dear Dr. Tan, 

I'm pleased to inform you that your manuscript has been deemed suitable for publication in PLOS ONE. Congratulations! Your manuscript is now being handed over to our production team.

Kind regards, 

on behalf of

Dr. Xiaobao Yu 

Academic Editor

PLOS ONE